# On the Plasticity and Stability for Post-Training Large Language Models

**Wenwen Qiang** [1 2]  **Ziyin Gu** [1 2]  **Jiahuan Zhou** [3]  **Jie Hu** [4]  **Jingyao Wang** [1 2]  **Changwen Zheng** [1 2]  **Hui Xiong** [5 6]

## Abstract

Training stability remains a critical bottleneck for Group Relative Policy Optimization (GRPO), often manifesting as a trade-off between reasoning plasticity and general capability retention. We identify a root cause as the geometric conflict between plasticity and stability gradients, which leads to destructive interference. Crucially, we argue that deterministic projection methods are suboptimal for GRPO as they overlook the intrinsic stochasticity of group-based gradient estimates. To address this, we propose **P**robabilistic **C**onflict **R**esolution (**PCR**), a Bayesian framework that models gradients as random variables. PCR dynamically arbitrates conflicts via an uncertainty-aware "soft projection" mechanism, optimizing the signal-to-noise ratio. Extensive experiments demonstrate that PCR significantly smooths the training trajectory and achieves superior performance in various reasoning tasks.

## 1. Introduction

Large Language Models (LLMs) like DeepSeek-R1 (Guo et al., 2025) has proven that Reinforcement Learning (RL) is essential for unlocking complex reasoning capabilities. Among the various RL techniques, Group Relative Policy Optimization (GRPO) (Shao et al., 2024) has become the standard choice. By removing the need for a separate value network, GRPO significantly reduces memory usage and enables scalable training, allowing models to learn effectively from group-based relative rewards.

However, despite its efficiency, GRPO is notoriously difficult to train (Zhang et al., 2026; Ge et al., 2025; Wu et al., 2025). Practitioners face a harsh "Plasticity-Stability Dilemma": aggressive updates to improve reasoning (plasticity) often cause the model to forget general knowledge or lose linguistic coherence (stability). Conversely, strict constraints to preserve language often prevent the model from learning new reasoning skills. Balancing these two forces typically requires exhaustive, fragile tuning of the KL penalty coefficient ($\beta$).

Why is GRPO so unstable? In this paper, inspired by (Liu et al., 2021), we argue that a possible root cause is a geometric conflict between two optimization objectives in the GRPO loss. Our analysis reveals that the gradient for reasoning ($\mathbf{g}_{pla}$) and the gradient for stability ($\mathbf{g}_{sta}$) frequently point in opposite directions. The standard GRPO update rule simply adds these two opposing vectors together. This results in "destructive interference", where the two forces cancel each other out, causing the optimizer to fight against itself and traverse the loss landscape inefficiently.

Addressing this conflict is not as simple as using geometric projection methods like PCGrad (Yu et al., 2020). These methods assume that the calculated gradients are perfect, deterministic vectors. However, in GRPO, gradients are Monte Carlo estimates derived from a small group of training queries, Therefore, they are inherently noisy and uncertain. If we blindly project one gradient vector onto another based on noisy data (a "hard projection"), we risk discarding valid learning signals or enforcing incorrect constraints. We need a method that considers not just the direction of the gradients, but our confidence in them.

To solve this, we propose **P**robabilistic **C**onflict **R**esolution (**PCR**). Instead of treating gradients as fixed arrows, we model them as probability distributions (Gaussian random variables) to capture their uncertainty. We then employ Bayesian inference to dynamically arbitrate the conflict. The core idea is intuitive: if the reasoning signal is strong and precise (low variance), PCR trusts it and allows the update; if the reasoning signal is noisy or the stability constraint is rigid (high variance), PCR suppresses the update. This results in a soft projection mechanism. Unlike PCGrad, which deletes conflicting components entirely, PCR scales

[1]Institute of Software Chinese Academy of Sciences, Beijing, China [2]University of the Chinese Academy of Sciences, Beijing, China [3]Wangxuan Institute of Computer Technology, Peking University, Beijing, China [4]Meituan [5]Thrust of Artificial Intelligence, The Hong Kong University of Science and Technology (Guangzhou), China [6]Department of Computer Science and Engineering, The Hong Kong University of Science and Technology Hong Kong SAR, China. Correspondence to: Jiahuan Zhou <jiahuanzhou@pku.edu.cn>, Jingyao Wang <wangjingyao2023@iscas.ac.cn>, Changwen Zheng <changwen@iscas.ac.cn>.

*Proceedings of the 43rd International Conference on Machine Learning*, Seoul, South Korea. PMLR 306, 2026. Copyright 2026 by the author(s).

them based on the signal-to-noise ratio. To make this computationally feasible for billion-parameter models, we apply PCR strategically: we use it only on the MLP layers (which act as knowledge stores) while using standard updates for Attention layers. This hybrid approach ensures rigorous stability for core knowledge without slowing down training.

Our contributions: **1**) We identify that the instability in GRPO stems from the high-dimensional geometric conflict between plasticity and stability gradients; **2**) We propose PCR, a Bayesian framework that derives a closed-form, uncertainty-aware "soft projection" rule. To our knowledge, this is the first work to introduce probabilistic modeling into gradient projection for LLM post-training; **3**) We introduce an efficient hybrid implementation that applies PCR only to MLP layers, making it scalable for LLM training; **4**) Theoretically, we prove that PCR is the mathematically optimal estimator in the gradient space. It minimizes the update error by finding the perfect trade-off between bias and variance. Empirically, extensive experiments demonstrate that this stability allows PCR to eliminate training oscillations and achieve superior performance on reasoning tasks.

## 2. Related Work

Large language models (LLMs) have achieved remarkable success in reasoning tasks through reinforcement learning post-training (Ouyang et al., 2022; Wang et al., 2026). While GRPO (Shao et al., 2024) has emerged as a dominant paradigm due to its memory efficiency, training stability remains a critical bottleneck. Recent studies have proposed various mechanisms to mitigate this volatility. On one hand, methods like $\Delta$L Norm (He et al., 2025) and BNPO (Xiao et al., 2025) introduce advanced normalization statistics to smooth the reward landscape and reduce variance. On the other hand, approaches like GSPO (Zheng et al., 2025) and BAPO (Xi et al., 2025) focus on refining policy constraints, employing adaptive clipping or sequence-level optimization to prevent policy divergence. Other works such as GVPO (Zhang et al., 2026) and MRT (Qu et al., 2025) utilize analytic re-weighting or reward correction. Meanwhile, GCPO (Gu et al., 2026) maintains stable training by constructing causal collision structures. Despite these diverse improvements, most existing methods focus on shaping the scalar loss or reward values. They often overlook the high-dimensional geometric antagonism between the plasticity gradient and the stability constraint, which our work identifies as a possible root cause of optimization conflict.

The challenge of conflicting gradients is a central theme in Multi-Task Learning (MTL). When objectives compete, simple summation leads to destructive interference. Classic solutions like PCGrad (Yu et al., 2020) project conflicting gradients onto the normal plane, while recent advances like MMPareto (Wei & Hu, 2024) and Robust MTL (He et al., 2024) explore Pareto-optimal frontiers and risk minimization to balance innocent unimodal assistance or excess risks. However, these methods typically operate under a deterministic assumption, treating gradient estimates as reliable ground truths. In the context of GRPO, gradients are Monte Carlo estimates carrying significant stochastic noise. Applying hard geometric projection to such noisy signals can erroneously discard valid exploration. Our proposed PCR bridges this gap by modeling gradients as probabilistic distributions. Instead of a hard cut, PCR introduces a Bayesian arbitration mechanism that performs "soft projection" based on the signal-to-noise ratio, offering a mathematically optimal trade-off for stochastic optimization.

## 3. Problem Formulation and Analysis

To rigorously analyze the plasticity-stability trade-off in post-training, we first reformulate the GRPO objective by decoupling it into two distinct forces. We then derive their respective gradients to examine their physical roles. Finally, through empirical analysis, we demonstrate that the inherent geometric conflict between these update directions serves as the root cause of training instability.

### 3.1. Reformulating the GRPO Objective

A LLM can be formulated as an autoregressive policy $\pi_\theta(\cdot \mid q)$, which generates a response token-by-token conditioned on a query $q$. GRPO optimizes this policy by introducing a group relative advantage. Given a query $q \sim P$ and a group of candidate outputs $\{y_i\}_{i=1}^n$ sampled from the old policy $\pi_{\theta_{\text{old}}}$, the standard GRPO objective aims to maximize:

$$\mathcal{J}_{\text{GRPO}}(\theta) = \mathbb{E}_{q \sim P, \{y_i\} \sim \pi_{\theta_{\text{old}}}} \big[ \frac{1}{n} \sum_{i=1}^n \frac{1}{T_i} \tag{1}$$
$$\sum_{j=1}^{T_i} \left( \mathcal{S}_{i,j}(\theta) - \beta \mathcal{K}_{i,j}(\theta) \right) \big].$$

To clearly identify the sources of the gradients, we formally define the two core components in the objective above:

(1) **Token-level Surrogate Gain ($\mathcal{S}_{i,j}$):** This term drives the model updates towards regions of higher reward. We define the importance ratio as $R_{i,j}(\theta) = \frac{\pi_\theta(y_{i,j}|q, y_{i,<j})}{\pi_{\theta_{\text{old}}}(y_{i,j}|q, y_{i,<j})}$. Incorporating the clipping mechanism for training stability, the surrogate gain is defined as:

$$\mathcal{S}_{i,j}(\theta) = \min \left( R_{i,j}(\theta) A_i, \ \text{clip}(R_{i,j}(\theta), 1 - \epsilon, 1 + \epsilon) A_i \right), \tag{2}$$

where $A_i$ denotes the group relative advantage, computed by standardizing the rewards within the group: $A_i = (r_i - \mu_{\text{group}})/\sigma_{\text{group}}$, and $y_i = (y_{i,1}, \cdots, y_{i,T_i})$.

(2) **Token-level KL Penalty ($\mathcal{K}_{i,j}$):** This term acts as a regularizer, constraining the policy from deviating excessively from the reference distribution:

$$\mathcal{K}_{i,j}(\theta) = D_{\text{KL}} \left( \pi_\theta(\cdot|q, y_{i,<j}) \parallel \pi_{\text{ref}}(\cdot|q, y_{i,<j}) \right). \tag{3}$$

Here, $\pi_{\text{ref}}$ is typically set to the old policy $\pi_{\theta_{\text{old}}}$.

## 3.2. Dual Decomposition of Loss and Gradients

For gradient analysis, we transform the maximization problem of $\mathcal{J}_{\text{GRPO}}$ into a minimization problem of a loss function $\mathcal{L}_{\text{GRPO}} = -\mathcal{J}_{\text{GRPO}}$. Based on the Eq. 1, we explicitly decompose the total loss into two independent terms representing plasticity and stability:

$$\mathcal{L}_{\text{GRPO}}(\theta) = \underbrace{\mathcal{L}_{\text{pla}}(\theta)}_{\text{Plasticity}} + \beta \cdot \underbrace{\mathcal{L}_{\text{sta}}(\theta)}_{\text{Stability}}, \qquad (4)$$

Here, the plasticity loss corresponds to the negative expectation of the surrogate gain. It aims to enhance task-specific performance by exploiting the advantage signals:

$$\mathcal{L}_{\text{pla}}(\theta) = -\mathbb{E}[\frac{1}{n}\sum_{i=1}^{n}\frac{1}{T_i}\sum_{j=1}^{T_i}\mathcal{S}_{i,j}(\theta)]. \qquad (5)$$

The stability loss corresponds to the KL divergence term. It aims to anchor the policy to the reference manifold to preserve general capabilities:

$$\mathcal{L}_{\text{sta}}(\theta) = \mathbb{E}[\frac{1}{n}\sum_{i=1}^{n}\frac{1}{T_i}\sum_{j=1}^{T_i}\mathcal{K}_{i,j}(\theta)]. \qquad (6)$$

From the above, we derive two gradient vectors used for parameter updates. Relying on the linearity of the gradient operator, the total gradient $\mathbf{g}_{total}$ can be expressed as:

$$\nabla_{\theta}\mathcal{L}_{\text{GRPO}}(\theta) = \mathbf{g}_{pla} + \beta \cdot \mathbf{g}_{sta}. \qquad (7)$$

Suppose a batch has $N_{\text{batch}}$ queries, these two gradients possess distinct physical interpretations and adversarial natures:

(1) **The Plasticity Gradient ($\mathbf{g}_{pla}$):**

$$\mathbf{g}_{pla} \triangleq \nabla_{\theta}\mathcal{L}_{\text{pla}}(\theta) \approx -\frac{1}{N_{\text{batch}}}\sum \nabla_{\theta}\mathcal{S}_{i,j}(\theta). \qquad (8)$$

This gradient represents the signal for policy improvement. It drives the parameters $\theta$ in a direction that maximizes the specific task reward, serving as the primary source of the model's new capabilities (i.e., plasticity).

(2) **The Stability Gradient ($\mathbf{g}_{sta}$):**

$$\mathbf{g}_{sta} \triangleq \nabla_{\theta}\mathcal{L}_{\text{sta}}(\theta) \approx \frac{1}{N_{\text{batch}}}\sum \nabla_{\theta}\mathcal{K}_{i,j}(\theta). \qquad (9)$$

This gradient represents the signal for behavior maintenance. It drives the parameters $\theta$ back towards the parameter space of the reference model, serving as the key constraint to prevent catastrophic forgetting and maintain general capabilities (i.e., stability).

## 3.3. Empirical Analysis

While GRPO theoretically balances policy improvement and reference maintenance, in practice, it exhibits significant training instability (Ouyang et al., 2022; Simoni et al., 2025; Dai et al., 2025). Models often oscillate between overfitting to the reasoning task (forgetting general knowledge) and stagnation (failing to learn reasoning), requiring exhaustive hyperparameter tuning. To diagnose the root cause of this instability, we analyze the optimization dynamics of DeepSeek-R1-Distill-Llama-8B on the AIME dataset (detailed setup in Appendix B).

We first visualize the trade-off between reasoning performance (AIME Pass@1) and linguistic stability (WikiText-2 PPL) by varying the KL coefficient $\beta$. The results are shown in Figure 1(a)-(b). We can observe that a slight decrease in $\beta$ triggers a disproportionate collapse in PPL (Note that a lower PPL is better), while a slight increase suppresses reasoning gains entirely. As shown in Figure 1(b), the Pareto frontier is extremely sharp. This sensitivity suggests that the two objectives are functionally antagonistic, and standard scalarization fails to find a stable equilibrium, leading to the observed training volatility.

To confirm this antagonism physically, we compute the layer-wise cosine similarity between the Plasticity gradient $\mathbf{g}_{pla}$ and the stability gradient $\mathbf{g}_{sta}$. The heatmap in Figure 1(b) reveals that the middle-to-deep MLP layers-critical for knowledge storage-exhibit persistent negative cosine similarity (gradient conflict) throughout training. This geometric conflict implies that the standard GRPO update rule ($\mathbf{g}_{pla} + \beta\mathbf{g}_{sta}$) results in destructive interference: the opposing vectors partially cancel each other out, reducing the effective update magnitude and skewing the direction. This cancellation explains the instability: the optimizer is fighting against itself, leading to inefficient traversal of the loss landscape and sensitivity to noise.

## 4. Methodology

In this section, we introduce Probabilistic Conflict Resolution (PCR), a principled framework that dynamically arbitrates gradient conflicts in GRPO. Our method unfolds in four key stages: probabilistic modeling, geometric decomposition, Bayesian arbitration, and finally reconstruction and projection. This pipeline effectively transforms the hard geometric constraints of prior methods into a flexible, uncertainty-aware optimization process.

### 4.1. Probabilistic Modeling

As formalized in the previous section, the plasticity gradient $\mathbf{g}_{pla}$ and the stability gradient $\mathbf{g}_{sta}$ frequently exhibit adversarial directions. The canonical approach to resolving such conflicts is PCGrad (Yu et al., 2020), which treats gradients

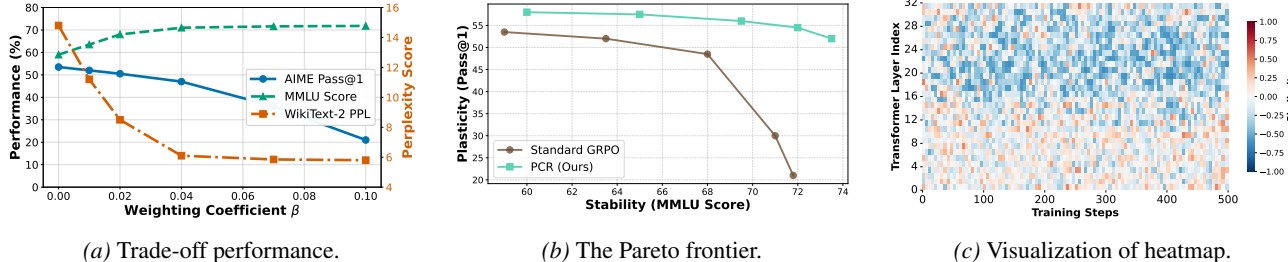

*(a)* Trade-off performance.    *(b)* The Pareto frontier.    *(c)* Visualization of heatmap.

*Figure 1.* Motivating results. (a) Results of AIME accuracy, MMLU score, and PPL, varying with the KL coefficient $\beta$. (b) The Pareto frontier. (c) The layer-wise cosine similarity between plasticity and stability gradients across training steps.

as deterministic vectors and performs geometric projection upon conflict. However, this deterministic perspective overlooks a fundamental property of GRPO optimization: the observed gradients are not true expectations over the full data distribution but are Monte Carlo estimators derived from a finite group of samples.

From a statistical lens, gradient estimates are often noisy. A gradient with high variance is unreliable, meaning we cannot fully trust its direction. In such cases, PCGrad blindly applies projection regardless of uncertainty, creating the risk of over-trusting a noisy direction and removing valid signals. Intuitively, the projection should be adaptive based on confidence. If the gradient is precise (low variance), we should strictly perform geometric projection. However, if the gradient is noisy and random (high variance), we should trust it less and perform a weaker projection to avoid being misled by noise. Therefore, to formalize this adaptive mechanism, we must model gradients as random variables rather than deterministic vectors, which allows us to quantify the uncertainty (refer to Appendix C for more explanation).

To mathematically capture this uncertainty, we look at the aggregation mechanism of GRPO. Since the gradient is computed by averaging over a group of independent stochastic queries $\{q_i\}_{i=1}^{N_{\text{batch}}}$, we can invoke the Central Limit Theorem (CLT) (Feller, 1991). The CLT states that the distribution of the sample mean approaches a multivariate Gaussian as the sample size implies. This provides a rigorous justification for approximating the gradient estimation as Gaussian, a standard practice in the analysis of stochastic optimization dynamics (Mandt et al., 2017). By leveraging this property, we can explicitly quantify the confidence of update directions via their covariance structure. Thus, we have:

$$\mathbf{g}_{pla} \sim \mathcal{N}(\boldsymbol{\mu}_{pla}, \Sigma_{pla}), \ \mathbf{g}_{sta} \sim \mathcal{N}(\boldsymbol{\mu}_{sta}, \Sigma_{sta}), \ (10)$$

where $\boldsymbol{\mu}_{pla}$ and $\boldsymbol{\mu}_{sta}$ represent the theoretical expectations of the gradients over the underlying data distribution, e.g., the latent ground truths we aim to approximate. Meanwhile, $\Sigma$ characterize the estimation uncertainty, quantifying how significantly the observed gradients derived from group sampling diverge from these true expectations.

To bridge the gap between theoretical rigor and practical feasibility, we adopt three physically motivated approximations. First, regarding the direction, we approximate the latent true mean $\boldsymbol{\mu}$ directly using the observed gradient $\mathbf{g}$. This relies on the fundamental premise of stochastic approximation (Robbins & Monro, 1951), treating the empirical gradient as an unbiased estimator of the population expectation. Second, following standard adaptive optimization practices (e.g., Adam (Kingma & Ba, 2015)), we simplify the complex covariance structure via an isotropic assumption ($\Sigma \approx \sigma^2 \mathbf{I}$), where the scalar variance is estimated by the trace of intra-group gradients. This effectively captures the global noise level without incurring memory overhead. Third, inspired by the mean-field assumption common in variational inference (Blei et al., 2017), we treat the estimation errors of the two gradients as conditionally independent. This is justified by their orthogonal sources of stochasticity: the variance in $\mathbf{g}_{pla}$ is primarily driven by the instability of discrete rewards, whereas the variance in $\mathbf{g}_{sta}$ arises from the probabilistic divergence of the token distribution.

### 4.2. Geometric Decomposition

While the probabilistic model measures gradient uncertainty, the actual interaction between the two gradient distribution is determined by their expected directions, represented by the means $\boldsymbol{\mu}_{pla}$ and $\boldsymbol{\mu}_{sta}$ (refer to Appendix E for more detailed explanation). To rigorously incorporate uncertainty into conflict resolution, we must first analyze the geometric relationship between these two vectors. The goal of this step is to physically isolate the specific component responsible for the directional opposition.

We formally define a gradient conflict as a situation where the two expected gradients point in opposing directions, satisfying $\boldsymbol{\mu}_{pla} \cdot \boldsymbol{\mu}_{sta} < 0$. This simply means that the plasticity objective is pushing the parameters in a direction that hurts the stability objective. To resolve this, mere identification is not enough. We must geometrically separate the "safe" part of the update from the "harmful" part.

To isolate the specific source of the conflict, we perform an orthogonal decomposition of the plasticity gradient $\boldsymbol{\mu}_{pla}$

using the stability gradient $\boldsymbol{\mu}_{sta}$ as the reference frame. We choose the stability gradient as the anchor because of the Alignment Hypothesis (Zhou et al., 2024). It suggests that the pre-trained knowledge is the foundation of the model's intelligence. Therefore, our primary goal is to learn new capabilities without breaking this foundation. Based on this logic, we decompose the plasticity gradient as follows:

$$\boldsymbol{\mu}_{pla} = \boldsymbol{\mu}_{pla}^{\perp} + \boldsymbol{\mu}_{pla}^{\parallel}. \tag{11}$$

This formula splits the update into two components with clear physical meanings. The first is the independent component ($\boldsymbol{\mu}_{pla}^{\perp}$), which is perpendicular to the stability gradient. Since it moves in a direction that has no effect on the stability objective, it is structurally safe and should be fully preserved. The second is the conflicting component ($\boldsymbol{\mu}_{pla}^{\parallel}$), which is the projection of the plasticity gradient onto the stability gradient. When a conflict happens, this component points directly against the constraint:

$$\boldsymbol{\mu}_{pla}^{\parallel} = \frac{\boldsymbol{\mu}_{pla} \cdot \boldsymbol{\mu}_{sta}}{\|\boldsymbol{\mu}_{sta}\|^2} \boldsymbol{\mu}_{sta}. \tag{12}$$

This component represents the specific force that attempts to violate the stability rule. Note that in this step, we only identify this conflicting force geometrically. The decision of whether to remove it or keep it depends on its uncertainty, which we will handle in the next section.

### 4.3. Bayesian Arbitration

Having isolated the conflicting component $\boldsymbol{\mu}_{pla}^{\parallel}$, we face the central decision of determining exactly how much of this conflicting update to retain. Traditional methods like PCGrad are too extreme because they bluntly discard the entire component, implicitly assuming that the stability constraint is perfect and allows for no error. To correct this bias, we use Bayesian inference to dynamically balance the two objectives based on their reliability. Crucially, although the parameter space of an LLM is high-dimensional, the gradient conflict is strictly localized along a single line defined by the stability gradient $\boldsymbol{\mu}_{sta}$. Any movement perpendicular to this axis is already safe. Therefore, we can simplify this complex optimization into a scalar estimation problem for a variable $x$, denoting the optimal move along conflict axis.

To estimate this optimal magnitude, we combine two distinct sources of information. We view the plasticity gradient as a noisy observation (the Likelihood). It pushes the model to move by a magnitude of $x_{obs} = \|\boldsymbol{\mu}_{pla}^{\parallel}\|$, but its reliability is inherently limited by its variance. If this driving signal is precise, we trust the observation and move. Simultaneously, we view the stability requirement as a prior belief (the Prior). Its goal is to act as an anchor, aiming to prevent deviation from the reference manifold. Mathematically, this creates a preference for the update to be zero. The variance of this

prior acts like a physical stiffness parameter. A small variance represents a rigid wall that forbids movement, while a large variance acts like a flexible elastic band that permits necessary exploration.

By combining the likelihood and the prior using Bayes' theorem, the optimal update $x^*$ emerges as the precision-weighted average of the two. We formalize this result as:

**Proposition 4.1** (Optimal Conflict Retention). *The optimal update magnitude $x^*$ is governed by the following:*

$$x^* = k \cdot x_{obs}, \quad where \quad k = \frac{\lambda_{pla}}{\lambda_{pla} + \lambda_{sta}}. \tag{13}$$

*Here, $\lambda = 1/\sigma^2$ denotes precision, and the scalar $k \in [0, 1]$ is defined as the retention coefficient.*

The proof is provided in Appendix F. The coefficient $k$ offers a very clear physical interpretation: it represents the relative confidence of the plasticity signal compared to the stability constraint. If the plasticity signal is far more reliable, $k$ approaches 1 and we fully keep the update. Conversely, if the stability constraint is far more certain, $k$ approaches 0 and we discard the update. This allows the algorithm to smoothly interpolate between exploring and restraining based on the instantaneous signal-to-noise ratio.

### 4.4. Reconstruction and Projection

With the retention coefficient $k$ derived from Bayesian arbitration, we now reconstruct the final gradient update. The logic is simple: we preserve the safe component entirely since it violates no constraints, but we only retain a fraction $k$ of the conflicting component. Mathematically, the final gradient is composed as $\mathbf{g}_{final} = \boldsymbol{\mu}_{pla}^{\perp} + k \cdot \boldsymbol{\mu}_{pla}^{\parallel}$. By using the geometric definition $\boldsymbol{\mu}_{pla}^{\perp} = \boldsymbol{\mu}_{pla} - \boldsymbol{\mu}_{pla}^{\parallel}$, we can rewrite this equation to show that the final gradient is simply the original gradient minus a correction term:

$$\mathbf{g}_{final} = (\boldsymbol{\mu}_{pla} - \boldsymbol{\mu}_{pla}^{\parallel}) + k \cdot \boldsymbol{\mu}_{pla}^{\parallel} = \boldsymbol{\mu}_{pla} - (1-k)\boldsymbol{\mu}_{pla}^{\parallel}. \tag{14}$$

To make this physically intuitive, we define a new term $\alpha = 1 - k$, which represents the projection strength (i.e., how much of the conflict we strictly remove). Substituting the precision terms derived earlier, $\alpha$ takes the form $\alpha = \lambda_{sta}/(\lambda_{pla} + \lambda_{sta})$. Finally, by plugging in the formula for the conflicting component, we arrive at the final update:

$$\boxed{\mathbf{g}_{final} = \boldsymbol{\mu}_{pla} - \alpha \frac{\boldsymbol{\mu}_{pla} \cdot \boldsymbol{\mu}_{sta}}{\|\boldsymbol{\mu}_{sta}\|^2} \boldsymbol{\mu}_{sta}} \tag{15}$$

This closed-form solution reveals that PCR is essentially a soft projection algorithm where the projection strength $\alpha$ is not manually set but automatically calculated from the data. We can validate its rationality by examining two extreme cases. First, consider the case of a high certainty

constraint where the stability gradient is very precise and rigid ($\lambda_{sta} \gg \lambda_{pla}$). Here, $\alpha$ approaches 1, meaning PCR converges to PCGrad and performs a hard projection to strictly eliminate the conflict. Conversely, consider the case of a low certainty constraint where the stability signal is noisy and unreliable ($\lambda_{sta} \ll \lambda_{pla}$). Here, $\alpha$ approaches 0, meaning PCR converges to standard gradient addition, effectively ignoring the constraint to allow full exploration. In this way, PCR achieves a smooth and theoretically grounded interpolation between obeying the constraint and ignoring it, based purely on which signal is more trustworthy.

### 4.5. Policy Optimization

While PCR provides a theoretically optimal solution for resolving conflicts, applying it element-wise to every single parameter in a billion-parameter LLM is computationally too expensive. Maintaining variance estimators and calculating projection coefficients for the entire model would drastically increase memory usage and slow down training. To make PCR practical for large-scale models, we introduce an efficient hybrid update strategy grounded in the specific roles of Transformer components.

Modern LLMs are built from alternating Self-Attention layers and Feed-Forward Networks (MLPs). Recent interpretability studies (Geva et al., 2021; Meng et al., 2022) offer a crucial insight: MLP layers act as factual knowledge stores, holding the vast majority of the model's domain expertise. In contrast, Attention layers serve primarily as context routers that move information around. Consequently, the problem of catastrophic forgetting, which represents the loss of model stability, is possibly caused by overwriting the knowledge stored within these MLP layers.

Based on this insight, we apply a strategic split. We use the computationally intensive PCR exclusively for the MLP layers to rigorously protect knowledge where it actually lives. For the Attention layers and other parameters (like Layer-Norm), we revert to the standard, efficient GRPO update (simple addition). This approach strikes an optimal balance: it secures the core knowledge modules without paying the computational price for the whole model. Mathematically, the update rule for parameters $\theta^{(l)}$ at layer $l$ is defined as:

$$\mathbf{g}_{update}^{(l)} = \begin{cases} \mathbf{g}_{final}^{(l)} \text{ (via Eq. 17)} & \text{if } \theta^{(l)} \in \text{MLP} \\ \mathbf{g}_{pla}^{(l)} + \beta\mathbf{g}_{sta}^{(l)} & \text{otherwise} \end{cases} \quad (16)$$

The parameters are then updated using the optimizer: $\theta_{t+1}^{(l)} \leftarrow \theta_t^{(l)} - \eta \cdot \text{Optimizer}(\mathbf{g}_{update}^{(l)})$. By restricting the overhead of PCR to only the most critical layers, we ensure training stability with negligible cost. The complete algorithm is detailed in Appendix D.

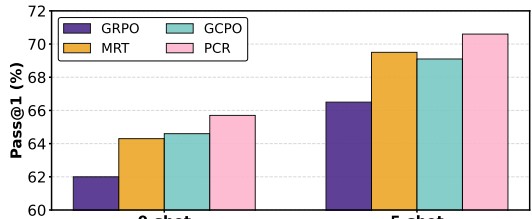

*Figure 2.* Performance analysis with PCR on code reasoning tasks. We record the 1-shot and 5-shot results on HumanEval.

## 5. Theoretical Analysis

In this section, we demonstrate that the projection coefficient derived in PCR is not a heuristic design but the precise analytical solution under the Minimum Mean Square Error (MMSE) criterion. This means that PCR mathematically finds the optimal balance between the bias caused by violating constraints and the variance caused by noisy gradients.

**Theorem 5.1** (MMSE Optimality of Soft Projection). *Consider the scalar estimation problem along the conflict axis defined by the unit vector $\mathbf{u} = -\boldsymbol{\mu}_{sta}/\|\boldsymbol{\mu}_{sta}\|$. Let the latent true update $z^* \in \mathbb{R}$ follow a prior distribution governed by the stability constraint $z^* \sim \mathcal{N}(0, \sigma_{sta}^2)$. Let the plasticity gradient provide a noisy observation $z_{obs} = z^* + \epsilon$, where the noise $\epsilon \sim \mathcal{N}(0, \sigma_{pla}^2)$ is independent of $z^*$. For the family of linear estimators $\hat{z}(\alpha) = (1 - \alpha)z_{obs}$ with $\alpha \in [0, 1]$, the projection coefficient $\alpha^* = \frac{\lambda_{sta}}{\lambda_{pla} + \lambda_{sta}}$ utilized by PCR achieves the global minimum of the posterior expected risk $\mathcal{R}(\alpha) = \mathbb{E}[(\hat{z}(\alpha) - z^*)^2]$.*

The proof is provided in Appendix G. Based on this theorem, we verify why PCR is superior to fixed strategies. By substituting $\alpha^*$ back into the risk function, the minimum risk of PCR is $\mathcal{R}_{PCR} = (\lambda_{pla} + \lambda_{sta})^{-1}$. In contrast, PCGrad (Hard Projection, $\alpha = 1$) yields a risk of $\sigma_{sta}^2$, while Naive Sum (No Projection, $\alpha = 0$) yields a risk of $\sigma_{pla}^2$. Since the combined precision is always higher than individual precisions, PCR is theoretically guaranteed to achieve lower error than both baselines, i.e., $\mathcal{R}_{PCR} < \mathcal{R}_{PCGrad}$ and $\mathcal{R}_{PCR} < \mathcal{R}_{Sum}$. The mechanism is physically intuitive. If the stability constraint is unreliable (large $\sigma_{sta}^2$), hard projection introduces a massive error known as bias. PCR avoids this by automatically reducing $\alpha$. Conversely, if the plasticity gradient is unstable (large $\sigma_{pla}^2$), a direct update introduces a massive error known as variance. PCR suppresses this noise by automatically increasing $\alpha$. Therefore, we can conclude that PCR guarantees better convergence stability than fixed strategy.

## 6. Experiment

In this section, we begin by describing the experimental setup. We then present the results of the evaluation, followed by an ablation study to analyze how it works well. More

*Table 1.* Pass@1 performance on various math reasoning benchmarks. We compare base models trained with different fine-tuning approaches. The best results are highlighted in **bold**. More details and results are provided in Appendix L.1.

| Base model + Method | AIME 2024 | AIME 2025 | AMC 2023 | MATH500 | MinervaMATH | Avg. |
|---|---|---|---|---|---|---|
| **DeepScaleR-1.5B-Preview** | 42.8 | 36.7 | 83.0 | 85.2 | 24.6 | 54.5 |
| GRPO (Shao et al., 2024) | 44.5 (+1.7) | 39.3 (+2.6) | 81.5 (-1.5) | 84.9 (-0.3) | 24.7 (+0.1) | 55.0 (+0.5) |
| GVPO (Zhang et al., 2026) | 46.1 (+3.3) | 39.7 (+3.0) | 83.6 (+0.6) | 85.7 (+0.5) | 25.3 (+0.7) | 56.1 (+1.6) |
| MRT (Qu et al., 2025) | 47.2 (+4.4) | 39.7 (+3.0) | 83.1 (+0.1) | 85.1 (-0.1) | 24.2 (-0.4) | 55.9 (+1.4) |
| GCPO (Gu et al., 2026) | 46.7 (+3.9) | 40.3 (+3.6) | 84.1 (+1.1) | 86.3 (+1.1) | 25.9 (+1.4) | 56.8 (+2.3) |
| PCR (Ours) | **48.1 (+5.3)** | **41.4 (+4.7)** | **84.9 (+1.9)** | **87.0 (+1.8)** | **26.8 (+2.2)** | **57.7 (+3.2)** |
| **DeepSeek-R1-Distill-Qwen-1.5B** | 28.7 | 26.0 | 69.9 | 80.1 | 19.8 | 44.9 |
| GRPO (Shao et al., 2024) | 29.8 (+1.1) | 27.3 (+1.3) | 70.5 (+0.6) | 80.3 (+0.2) | 22.1 (+2.3) | 46.0 (+1.1) |
| GVPO (Zhang et al., 2026) | 30.6 (+1.9) | 28.2 (+2.2) | 71.5 (+1.6) | 80.5 (+0.4) | 23.1 (+3.3) | 46.7 (+1.8) |
| MRT (Qu et al., 2025) | 30.3 (+1.6) | 29.3 (+3.3) | 72.9 (+3.0) | 80.4 (+0.3) | 22.5 (+2.7) | 47.1 (+2.2) |
| GCPO (Gu et al., 2026) | 31.0 (+2.3) | 29.0 (+3.0) | 71.8 (+1.9) | 81.6 (+1.5) | 23.4 (+3.6) | 47.4 (+2.5) |
| PCR (Ours) | **32.3 (+3.6)** | **30.4 (+4.4)** | **72.7 (+2.8)** | **82.2 (+2.1)** | **24.8 (+5.0)** | **48.5 (+3.6)** |
| **DeepSeek-R1-Distill-Qwen-7B** | 55.5 | 50.2 | 85.1 | 87.4 | 42.1 | 64.1 |
| GRPO (Shao et al., 2024) | 56.9 (+1.4) | 51.7 (+1.5) | 85.5 (+0.4) | 87.7 (+0.3) | 43.5 (+1.4) | 65.1 (+1.0) |
| GVPO (Zhang et al., 2026) | 57.5 (+2.0) | 52.1 (+1.9) | 86.3 (+1.2) | 88.5 (+1.1) | 44.2 (+2.1) | 65.7 (+1.6) |
| MRT (Qu et al., 2025) | 57.0 (+1.5) | 52.4 (+2.2) | 86.0 (+0.9) | 88.4 (+1.0) | 44.3 (+2.2) | 65.6 (+1.5) |
| GCPO (Gu et al., 2026) | 58.3 (+2.8) | 53.0 (+2.8) | 87.3 (+2.2) | 89.1 (+1.7) | 45.0 (+2.9) | 66.5 (+2.4) |
| PCR (Ours) | **59.7 (+4.2)** | **54.1 (+3.9)** | **88.0 (+2.9)** | **89.8 (+2.4)** | **46.5 (+4.4)** | **67.6 (+3.5)** |

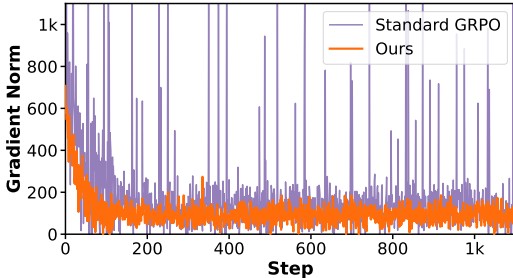

*Figure 3.* Stability analyses. We provide the norm of the gradient during training. A stable gradient norm implies consistent updates; large swings suggest unstable or overly aggressive shifts.

details and results are provided in the Appendices J-L.

## 6.1. Experimental Settings

We evaluate our method on a diverse suite of reasoning benchmarks, covering competition math and code generation, including AIME24–25, AMC, MATH500 (Hendrycks et al., 2021), MinervaMATH (Lewkowycz et al., 2022), and HumanEval (Chen et al., 2021). The experiments are conducted with four base models: DeepScaleR-1.5B-Preview, DeepSeek-R1-Distill-Qwen-1.5B, DeepSeek-R1-Distill-Qwen-7B, and Qwen2-7B-Instruct. We compare against representative classic and state-of-the-art (SOTA) RL post-training baselines, including GRPO (Shao et al., 2024), GVPO (Zhang et al., 2026), MRT (Qu et al., 2025), and GCPO (Gu et al., 2026). For data preparation, DeepScaleR-1.5B-Preview, which was previously fine-tuned on 40k math QA pairs, is further trained on 919 AIME problems. DeepSeek-R1-Distill-Qwen-1.5B is fine-tuned on a random subset of 4,000 QA pairs from NuminaMath (Li

et al., 2024). Following (Wang et al., 2026), we cap both training and evaluation with a token budget of 16,384. Unless otherwise noted, we use a learning rate of $1e-6$, weight decay of 0.01, and batch size of 256. All experiments are run on A100 GPU clusters. The code is available at CODE.

## 6.2. Performance Analysis

We evaluate our method, PCR, against all the mentioned baselines across diverse benchmarks. The evaluation involves three widely adopted base models, including DeepScaleR-1.5B-Preview, DeepSeek-R1-Distill-Qwen-1.5B, and DeepSeek-R1-Distill-Qwen-7B, with performance measured by pass@1 accuracy following the protocols in (Gu et al., 2026). The results are shown in Table 1. Across all experimental settings, our method consistently secures the top performance, surpassing both the base models and competitive RL baselines. Specifically, our method yields substantial improvements over the base models, achieving average gains of 3% on all the base models. Critical to this success is our Bayesian formulation, which effectively integrates generalizability and transferability. Unlike standard RL baselines that may overfit to specific reward signals, e.g., GRPO with negative performance gains in AMC, our method treats the optimization process as a Bayesian inference problem, i.e., our method demonstrates superior robustness on challenging benchmarks such as AIME and MinervaMATH, where it outperforms the strongest baselines by over nearly 1.2%. These results demonstrate the effectiveness of our method.

To further validate the versatility of PCR, we conduct experiments on code reasoning tasks using Qwen2-7B-Instruct.

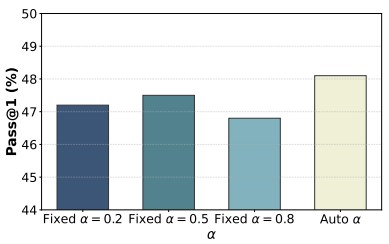

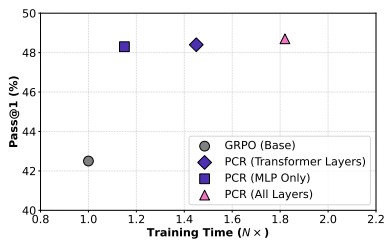

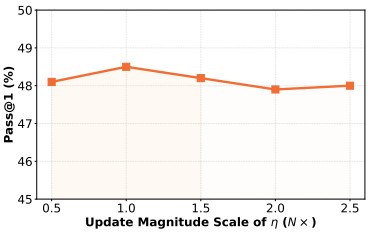

*(a)* Effect of probabilistic modeling.

*(b)* Effect of hybrid strategy.

*(c)* Impact of Learning Rate $\eta$.

*Figure 4.* (a) and (b) evaluate the effect of different components within PCR. (c) shows the scalability to larger update magnitudes.

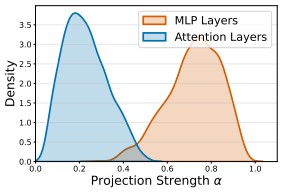

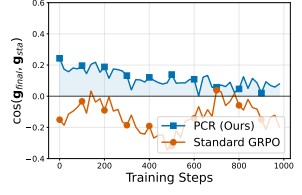

*Figure 5.* Visualization results. (a) The distribution of projection strength. (b) The cosine similarity between $\mathbf{g}_{final}$ and $\mathbf{g}_{sta}$.

We evaluate performance via the standard HumanEval protocol under both 0-shot and 5-shot settings. The results are presented in Figure 2. From the results, we can observe that our method achieves the best results among all methods: PCR achieves the improvements of almost 1.2% over the baselines. This further shows the advantages of our method.

### 6.3. Visualization Analysis

To further investigate how PCR dynamically resolves gradient conflicts, we conduct a series of visualization experiments. We first visualize the statistical distribution of the projection strength $\alpha$ across different functional modules (Figure 5(a)). Unlike PCGrad, which applies a binary projection (i.e., $\alpha \in \{0, 1\}$), PCR exhibits a continuous and adaptive distribution. Notably, we observe that MLP layers generally maintain a higher $\alpha$ density compared to Attention layers. This confirms our hypothesis that knowledge-heavy MLP modules encounter more severe directional conflicts, requiring stronger Bayesian arbitration to safeguard linguistic stability. We then measure the cosine similarity between the final update gradient $\mathbf{g}_{final}$ and the stability gradient $\mathbf{g}_{sta}$ throughout the post-training phase. As shown in Figure 5(b), PCR maintains a non-negative or slightly positive correlation. Besides, we also follow the same experimental settings in Subsection 3.3, and record the corresponding results. The results in Figure 6 shows that our method breaks the suboptimal Pareto frontier constrained by standard GRPO: PCR achieves superior AIME accuracy improvement while simultaneously maintaining WikiText-2 PPL. These results demonstrate that PCR successfully navigates the underlying conflicts, improving the LLM performance.

### 6.4. Training Stability Analyses

Given the high cost of post-training LLMs, optimization stability is essential for efficient convergence and avoiding collapse. Instability often stems from gradient conflict between reward maximization and reference preservation, which amplifies update variance and wastes compute. To quantitatively assess this, we utilize the gradient norm as a proxy for optimization smoothness, consistent with standard RL practices (Xiao et al., 2025). As shown in Figure 3, PCR exhibits the most stable training dynamics among all compared methods. While GRPO suffers from pronounced oscillations due to the conflicting gradients, PCR maintains a relatively smooth and consistent gradient norm throughout the training process. This superior stability demonstrates the effectiveness of our Bayesian fusion mechanism.

### 6.5. Ablation Studies

**The effect of different components within PCR.** We conduct ablation studies to justify the necessity of each component in PCR. Specifically, we first compare PCR (i.e., auto $\alpha$) against fixed soft projection baselines (i.e., $\alpha \in \{0.2, 0.5, 0.8\}$). The results in Figure 4(a) demonstrate that our method consistently outperforms any fixed heuristic, proving the superiority of Bayesian arbitration. Next, we evaluate applying PCR to different layer subsets. Our results in Figure 4(b) show that applying PCR exclusively to MLP layers achieves a performance gain comparable to the all layers setting while significantly reducing training time. This demonstrate the effectiveness of our design. To further isolate the contributions of uncertainty estimation, the Bayesian formulation, and the soft projection rule, we conduct a component-wise ablation study detailed in Appendix L.5. The results confirm that our closed-form Bayesian synthesis is mathematically optimal and empirically superior to other variations.

**Parameter Sensitivity.** We also conduct sensitivity studies on the weighting coefficient $\beta$ and the PCR parameter learning rate $\eta$. Figures 6a and 4c show that PCR is highly robust to variations in $\beta$ and continues to improve performance as the parameter update magnitude increases.

# 7. Conclusion

This work attributes the instability of GRPO to the high-dimensional conflict between task-specific optimization and reference maintenance. We argue that standard gradient aggregation fails due to the stochastic nature of group sampling. To resolve this, we introduced Probabilistic Conflict Resolution (PCR), which reformulates gradient projection as a Bayesian inference problem. By adaptively balancing exploration bias against constraint variance, PCR acts as an optimal linear filter in the gradient space. Our results confirm the effectiveness of the proposed PCR.

## Acknowledgments

The authors sincerely thank the anonymous reviewers for their valuable comments and constructive feedback. This work was supported in part by the National Natural Science Foundation of China under Grants No. 62506355, No. 62376011, and No. 92370204; in part by the National Key R&D Program of China under Grants No. 2024YFA1410000 and No. 2023YFF0725001; in part by the Guangdong Basic and Applied Basic Research Foundation under Grant No. 2023B1515120057; and in part by the Key-Area Special Project of Guangdong Provincial Ordinary Universities under Grant No. 2024ZDZX1007. This work was also recognized as a research achievement of the National Engineering Research Center of New Electronic Publishing Technologies.

## Impact Statement

This paper presents work whose goal is to advance the field of machine learning. There are many potential societal consequences of our work, none of which we feel must be specifically highlighted here.

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

# Appendix

The appendix of our paper is organized as follows:

- **Appendix A** provides the list of notations.

- **Appendix B** provides the experimental details for empirical analyses.

- **Appendix C** provides the conceptual analyses.

- **Appendix D** provides the pseudo code of the proposed PCR.

- **Appendix F** provides the proof of proposition.

- **Appendix G** provides the proof of theorem.

- **Appendix H** provides more comparisons.

- **Appendix I** illustrates the broader impacts and limitations.

- **Appendix K** provides the additional details of the benchmark datasets.

- **Appendix J** provides the additional details of the implementation details.

- **Appendix L** provides the full results and additional experiments.

## A. List of Notations

We list the definitions of all notations from the main text as follows:

☐ **Symbols of Problem Formulation and GRPO Objective**

- $q \sim P$: a query sampled from the distribution $P$.
- $\pi_\theta$: the policy (i.e., LLM) parameterized by $\theta$.
- $\pi_{\theta_{\mathrm{old}}}$: the old policy used to sample candidate outputs.
- $\pi_{\mathrm{ref}}$: the reference policy.
- $\{y_i\}_{i=1}^n$: a group of candidate outputs sampled from $\pi_{\theta_{\mathrm{old}}}$.
- $T_i$: the number of tokens in the response $y_i$.
- $\mathcal{J}_{\mathrm{GRPO}}(\theta)$: the standard GRPO optimization objective.
- $\mathcal{S}_{i,j}(\theta)$: the token-level surrogate gain.
- $\mathcal{K}_{i,j}(\theta)$: the token-level KL penalty.
- $A_i$: the group relative advantage.
- $\epsilon$: the clipping parameter for the surrogate gain.
- $\beta$: the coefficient for the KL penalty term.

☐ **Symbols of Loss Decomposition and Gradients**

- $\mathcal{L}_{\mathrm{GRPO}}(\theta)$: the total loss function for GRPO.
- $\mathcal{L}_{\mathrm{pla}}(\theta)$: the plasticity loss, corresponding to the negative expectation of the surrogate gain.
- $\mathcal{L}_{\mathrm{sta}}(\theta)$: the stability loss, corresponding to the KL divergence term.
- $\mathbf{g}_{pla}$: the plasticity gradient vector ($\nabla_\theta \mathcal{L}_{\mathrm{pla}}$).
- $\mathbf{g}_{sta}$: the stability gradient vector ($\nabla_\theta \mathcal{L}_{\mathrm{sta}}$).
- $N_{\mathrm{batch}}$: the number of queries in a batch.

☐ **Symbols of PCR**

- $\boldsymbol{\mu}_{pla}, \boldsymbol{\mu}_{sta}$: the theoretical expectations of the plasticity and stability gradients.

- $\Sigma_{pla}, \Sigma_{sta}$: the covariance matrices characterizing the estimation uncertainty of the gradients.
- $\sigma^2$: the scalar approximation of covariance estimated via trace variance.
- $\boldsymbol{\mu}_{pla}^{\perp}$: the independent component of the plasticity gradient orthogonal to the stability gradient.
- $\boldsymbol{\mu}_{pla}^{\parallel}$: the conflicting component of the plasticity gradient projected onto the stability gradient.
- $x$: the latent optimal update magnitude along the conflict axis.
- $x_{obs}$: the observed magnitude of the conflicting component, serving as a noisy measurement.
- $x^*$: the MAP estimate of the update magnitude.
- $\lambda_{pla}, \lambda_{sta}$: the precision of the plasticity and stability gradients.
- $k$: the retention coefficient representing the informational dominance of the plasticity task.
- $\mathbf{g}_{final}$: the final calibrated gradient update vector.

# B. Experimental Details for Empirical Analysis

### B.1. Experimental Setup

To verify the gradient conflict hypothesis presented in Section 3.3, we utilized the following setup:

- **Model & Dataset:** We used `DeepSeek-R1-Distill-Llama-8B` as the backbone. The training set consists of the AIME 2024 dataset, chosen for its high difficulty which forces the model to explore reasoning paths far from the pre-trained manifold.

- **Training Protocol:** We followed the standard GRPO protocol (Shao et al., 2024) with full-parameter fine-tuning to capture the global gradient landscape. The reference policy $\pi_{\mathrm{ref}}$ was initialized from the official checkpoint.

- **Metrics:**
  - *Plasticity:* Pass@1 accuracy on the AIME validation set.
  - *Stability:* Perplexity (PPL) on the WikiText-2 test set (Merity et al., 2016) and 5-shot accuracy on MMLU (Hendrycks et al., 2020).

- **Gradient Similarity Calculation:** We computed the cosine similarity at each layer $l$ and step $t$ as:

$$\mathrm{Sim}_t^{(l)} = \frac{\mathbf{g}_{pla}^{(l)} \cdot \mathbf{g}_{sta}^{(l)}}{\|\mathbf{g}_{pla}^{(l)}\|\|\mathbf{g}_{sta}^{(l)}\| + \epsilon} \tag{17}$$

  where $\mathbf{g}_{pla}^{(l)}$ and $\mathbf{g}_{sta}^{(l)}$ are the aggregated gradients for the plasticity loss and KL penalty, respectively.

### B.2. Additional Observations

(You can add more detailed plots or analysis here if needed, e.g., comparison with LoRA, or layer-wise breakdown.)

# C. Conceptual Analysis: Uncertainty-Aware Projection

To provide a deeper intuitive understanding of the motivations presented in Section 4, this appendix elaborates on the definitions of variance, confidence, and uncertainty within the context of gradient optimization, and illustrates how these statistical properties fundamentally dictate the optimal projection strategy.

### C.1. Definitions and Relationships

In the deterministic view (e.g., standard SGD or PCGrad), a gradient is treated as a single, infinitely thin arrow pointing in a specific direction. However, in the stochastic regime of GRPO, this simplification is inadequate. We define the core concepts as follows:

- **Variance ($\sigma^2$):** Mathematically, this measures the spread or dispersion of the gradient samples within a group. Physically, it represents the *noise level* of the signal. A high variance implies that the individual samples disagree significantly about the update direction.

- **Uncertainty:** This is the qualitative interpretation of variance. It reflects our *lack of knowledge* about the true underlying gradient. High variance directly translates to high uncertainty. If the gradient is uncertain, the "arrow" is not a sharp vector but rather a diffuse **probability cloud**.

- **Confidence:** This is the inverse of uncertainty. It represents the *trustworthiness* of the estimated direction. A signal is "confident" only when it is consistent across samples (i.e., low variance).

### C.2. Visualizing the Impact on Projection

How do these concepts influence the conflict resolution (projection) process? Consider the stability gradient not as a line, but as a boundary that defines the "safe zone."

**Scenario 1: High Confidence (Low Variance) → The "Solid Wall"**    When the stability gradient has low variance, the estimation is precise.

- **Visual Intuition:** The gradient vector is a sharp, thin arrow. The boundary it defines is a crisp, concrete line, acting like a **solid brick wall**.

- **Projection Logic:** Since we are confident that crossing this line violates the stability constraint, the projection must be **strict**. We should perform a "hard projection" (like PCGrad) to completely eliminate any component pointing towards the wall.

**Scenario 2: Low Confidence (High Variance) → The "Foggy Zone"**    When the stability gradient has high variance, the estimation is dominated by noise.

- **Visual Intuition:** The gradient vector is a diffuse, wide cone. The boundary it defines is not a line, but a **blurry, wide region of fog**. We do not know exactly where the safe zone ends and the violation begins.

- **Projection Logic:** If we treat this fog as a solid wall (blind projection), we might stop the model from exploring valid regions simply because of random noise. Therefore, the projection should be **weak**. We should only gently guide the update, or effectively ignore the constraint, allowing the model to pass through the fog to extract potential rewards.

### C.3. Summary

In summary, the transition from "hard projection" to "soft projection" is not an arbitrary heuristic but a necessary adaptation to the information content of the gradients. By modeling gradients as random variables, we convert the rigid geometric operation into a **signal-to-noise ratio arbitration**: we obey the signal (confident constraint) and ignore the noise (uncertain constraint).

## D. Pseudo Code of the proposed PCR

The complete training procedure is shown in the following:

## E. Theoretical Justification for Mean-Based Geometric Decomposition

In the methodology section, we perform geometric decomposition based on the expected gradients ($\boldsymbol{\mu}_{pla}$ and $\boldsymbol{\mu}_{sta}$) rather than operating on the raw stochastic samples or the full distributions directly. A natural question arises: *Does focusing on the means neglect the distributional nature of the gradients?*

In this section, we prove that under the assumption of linear projection, decomposing the means is mathematically equivalent to decomposing the entire distribution in expectation. Furthermore, we argue from the perspective of Stochastic Approximation theory that the "conflict" is a property of the underlying loss landscape (Signal), whereas the deviation is a property of sampling (Noise).

---

**Algorithm 1** Probabilistic Conflict Resolution (PCR) with Hybrid Update

---

1: **Input:** Policy $\pi_\theta$, Reference $\pi_{\text{ref}}$, Group Size $n$, Learning Rate $\eta$
2: **while** not converged **do**
3:     Sample query $q \sim P(q)$
4:     Generate group outputs $\{y_1, \ldots, y_n\} \sim \pi_\theta(\cdot|q)$
5:     **for** each layer $l$ **do**
6:         Compute gradients $\mathbf{g}_{pla}^{(l)}$ and $\mathbf{g}_{sta}^{(l)}$ via Eq. (8) and Eq. (9)
7:         **if** $\theta^{(l)} \in$ MLP **then**
8:             Estimate variances $\sigma_{pla}^2, \sigma_{sta}^2$ from group samples
9:             Compute means $\boldsymbol{\mu}_{pla}, \boldsymbol{\mu}_{sta}$
10:            **if** $\boldsymbol{\mu}_{pla} \cdot \boldsymbol{\mu}_{sta} < 0$ **then**
11:               Compute projection $\alpha$ via Eq. (16)
12:               $\mathbf{g}_{final}^{(l)} \leftarrow \boldsymbol{\mu}_{pla} - \alpha \frac{\boldsymbol{\mu}_{pla} \cdot \boldsymbol{\mu}_{sta}}{\|\boldsymbol{\mu}_{sta}\|^2} \boldsymbol{\mu}_{sta}$
13:            **else**
14:               $\mathbf{g}_{final}^{(l)} \leftarrow \boldsymbol{\mu}_{pla} + \beta \boldsymbol{\mu}_{sta}$
15:            **end if**
16:         **else**
17:            $\mathbf{g}_{final}^{(l)} \leftarrow \mathbf{g}_{pla}^{(l)} + \beta \mathbf{g}_{sta}^{(l)}$
18:         **end if**
19:         Update parameters: $\theta^{(l)} \leftarrow \text{Optimizer}(\theta^{(l)}, \mathbf{g}_{final}^{(l)})$
20:     **end for**
21: **end while**

---

## E.1. Commutativity of Linear Projection and Expectation

Let the plasticity gradient be a random variable $\mathbf{g}_{pla} \sim \mathcal{D}(\boldsymbol{\mu}_{pla}, \Sigma_{pla})$. We define the orthogonal projection operator $\mathcal{P}_{\mathbf{v}}(\cdot)$ which projects a vector onto a reference vector $\mathbf{v}$:

$$\mathcal{P}_{\mathbf{v}}(\mathbf{x}) = \frac{\mathbf{x} \cdot \mathbf{v}}{\|\mathbf{v}\|^2} \mathbf{v}. \tag{18}$$

In our context, the reference vector is the stability gradient expectation $\boldsymbol{\mu}_{sta}$. The **Conflicting Component** of the random variable $\mathbf{g}_{pla}$, denoted as $\mathbf{g}_{pla}^{\parallel}$, is therefore also a random variable:

$$\mathbf{g}_{pla}^{\parallel} = \mathcal{P}_{\boldsymbol{\mu}_{sta}}(\mathbf{g}_{pla}). \tag{19}$$

We are interested in the expected behavior of this conflicting component. By applying the linearity of expectation, we derive:

$$\mathbb{E}[\mathbf{g}_{pla}^{\parallel}] = \mathbb{E}\left[\frac{\mathbf{g}_{pla} \cdot \boldsymbol{\mu}_{sta}}{\|\boldsymbol{\mu}_{sta}\|^2} \boldsymbol{\mu}_{sta}\right] = \frac{\mathbb{E}[\mathbf{g}_{pla}] \cdot \boldsymbol{\mu}_{sta}}{\|\boldsymbol{\mu}_{sta}\|^2} \boldsymbol{\mu}_{sta} = \frac{\boldsymbol{\mu}_{pla} \cdot \boldsymbol{\mu}_{sta}}{\|\boldsymbol{\mu}_{sta}\|^2} \boldsymbol{\mu}_{sta} = \boldsymbol{\mu}_{pla}^{\parallel}. \tag{20}$$

**Conclusion:** The expectation of the projected distribution is exactly the projection of the mean. This equality implies that analyzing the geometry of the means ($\boldsymbol{\mu}_{pla}$) captures the unbiased central tendency of the conflict. By decomposing the mean, we are effectively isolating the *systematic* component of the conflict. The *stochastic* component (the variance $\Sigma_{pla}$) does not disappear; it is carried over into the Bayesian arbitration step (Section 3.3), where it determines the retention coefficient $k$. Thus, our framework strictly separates the *geometric direction* (determined by means) from the *statistical confidence* (determined by variances).

## E.2. Signal-to-Noise Separation in Optimization

From the perspective of optimization theory, specifically the Robbins-Monro framework (Robbins & Monro, 1951), a stochastic gradient $\mathbf{g}$ can be decomposed into the true gradient (Signal) and a zero-mean noise term (Noise):

$$\mathbf{g} = \nabla \mathcal{J}(\theta) + \boldsymbol{\epsilon}, \quad \text{where } \mathbb{E}[\boldsymbol{\epsilon}] = 0. \tag{21}$$

Here, $\boldsymbol{\mu} = \nabla \mathcal{J}(\theta)$ represents the true topological structure of the loss landscape.

- **True Conflict (Signal):** A conflict exists if and only if the true gradients oppose each other, i.e., $\nabla \mathcal{J}_{pla} \cdot \nabla \mathcal{J}_{sta} < 0$. This indicates that improving one objective structurally degrades the other.

- **Apparent Conflict (Noise):** Even if the true gradients are aligned ($\boldsymbol{\mu}_{pla} \cdot \boldsymbol{\mu}_{sta} > 0$), stochastic noise $\epsilon$ might cause a specific batch sample to exhibit negative cosine similarity. This is a sampling artifact, not a structural conflict.

If we were to determine conflict based on individual samples or the full distribution's overlap, we would risk over-correcting for noise-induced "pseudo-conflicts". By grounding the geometric decomposition on the means ($\boldsymbol{\mu}$), PCR targets the **structural antagonism** between the tasks while ignoring the ephemeral sampling noise. The uncertainty of the mean estimates is then handled by the subsequent Bayesian filtering, ensuring a robust and theoretically sound resolution.

## F. Proof of Proposition

**Proposition F.1** (Optimal Conflict Retention). *The optimal update magnitude $x^*$ is governed by the following:*

$$x^* = k \cdot x_{obs}, \quad where \quad k = \frac{\lambda_{pla}}{\lambda_{pla} + \lambda_{sta}}. \tag{22}$$

*Here, $\lambda = 1/\sigma^2$ denotes precision, and the scalar $k \in [0, 1]$ is defined as the retention coefficient.*

*Proof.* Here we provide the formal derivation for the Optimal Conflict Retention proposition. We formulate the conflict resolution as a Bayesian inference problem for a scalar latent variable $x$, representing the true optimal update along the conflict axis defined by $-\boldsymbol{\mu}_{sta}$.

We define the likelihood and prior as follows:

- **Likelihood:** $p(x_{obs}|x) = \mathcal{N}(x_{obs}|x, \sigma_{pla}^2) \propto \exp\left(-\frac{(x_{obs}-x)^2}{2\sigma_{pla}^2}\right)$

- **Prior:** $p(x) = \mathcal{N}(x|0, \sigma_{sta}^2) \propto \exp\left(-\frac{x^2}{2\sigma_{sta}^2}\right)$

The Maximum A Posteriori (MAP) estimate maximizes the posterior $p(x|x_{obs}) \propto p(x_{obs}|x)p(x)$. This is equivalent to minimizing the negative log-posterior loss function $\mathcal{L}(x)$:

$$\mathcal{L}(x) = \frac{(x - x_{obs})^2}{2\sigma_{pla}^2} + \frac{x^2}{2\sigma_{sta}^2} \tag{23}$$

Using precision notation $\lambda = 1/\sigma^2$, the stationarity condition $\frac{d\mathcal{L}}{dx} = 0$ yields:

$$\lambda_{pla}(x - x_{obs}) + \lambda_{sta}x = 0 \tag{24}$$

Solving for $x$:

$$(\lambda_{pla} + \lambda_{sta})x = \lambda_{pla}x_{obs} \implies x^* = \frac{\lambda_{pla}}{\lambda_{pla} + \lambda_{sta}}x_{obs} \tag{25}$$

Defining $k = \frac{\lambda_{pla}}{\lambda_{pla}+\lambda_{sta}}$, we obtain $x^* = k \cdot x_{obs}$. $\qquad\square$

## G. Proof of Theorem

**Theorem G.1** (MMSE Optimality of Soft Projection). *Consider the scalar estimation problem along the conflict axis defined by the unit vector $\mathbf{u} = -\boldsymbol{\mu}_{sta}/\|\boldsymbol{\mu}_{sta}\|$. Let the latent true update $z^* \in \mathbb{R}$ follow a prior distribution governed by the stability constraint $z^* \sim \mathcal{N}(0, \sigma_{sta}^2)$. Let the plasticity gradient provide a noisy observation $z_{obs} = z^* + \epsilon$, where the noise $\epsilon \sim \mathcal{N}(0, \sigma_{pla}^2)$ is independent of $z^*$. For the family of parameterized linear estimators $\hat{z}(\alpha) = (1 - \alpha)z_{obs}$ with $\alpha \in [0, 1]$, the projection coefficient $\alpha^* = \frac{\lambda_{sta}}{\lambda_{pla}+\lambda_{sta}}$ utilized by PCR achieves the global minimum of the posterior expected risk $\mathcal{R}(\alpha) = \mathbb{E}[(\hat{z}(\alpha) - z^*)^2]$.*

*Proof.* We expand the expected risk function $\mathcal{R}(\alpha)$ and utilize the structural property $z_{obs} = z^* + \epsilon$ to decompose it into Bias and Variance terms:

$$
\begin{aligned}
\mathcal{R}(\alpha) &= \mathbb{E}_{z^*, \epsilon}\left[((1-\alpha)(z^* + \epsilon) - z^*)^2\right] \\
&= \mathbb{E}_{z^*, \epsilon}\left[(-\alpha z^* + (1-\alpha)\epsilon)^2\right] \\
&= \underbrace{\alpha^2 \mathbb{E}[(z^*)^2]}_{\text{Bias}^2:\text{Constraint Rigidity}} + \underbrace{(1-\alpha)^2 \mathbb{E}[\epsilon^2]}_{\text{Variance: plasticity Noise}} + \underbrace{2\alpha(1-\alpha)\mathbb{E}[-z^* \epsilon]}_{\text{Cross Term}}.
\end{aligned}
\tag{26}
$$

By the independence assumption, the cross term vanishes: $\mathbb{E}[-z^* \epsilon] = -\mathbb{E}[z^*]\mathbb{E}[\epsilon] = 0$. Substituting the prior variance $\mathbb{E}[(z^*)^2] = \sigma_{sta}^2$ and the observation noise variance $\mathbb{E}[\epsilon^2] = \sigma_{pla}^2$, the risk function simplifies to a convex quadratic form with respect to $\alpha$:

$$
\mathcal{R}(\alpha) = \alpha^2 \sigma_{sta}^2 + (1-\alpha)^2 \sigma_{pla}^2.
\tag{27}
$$

To find the optimal $\alpha^*$, we set the derivative $\frac{d\mathcal{R}}{d\alpha} = 0$:

$$
2\alpha \sigma_{sta}^2 - 2(1-\alpha)\sigma_{pla}^2 = 0 \implies \alpha^* = \frac{\sigma_{pla}^2}{\sigma_{pla}^2 + \sigma_{sta}^2}.
\tag{28}
$$

Introducing the precision notation $\lambda = 1/\sigma^2$, we obtain $\alpha^* = \frac{\lambda_{sta}}{\lambda_{pla} + \lambda_{sta}}$. This is strictly identical to the definition in PCR. □

## H. More Comparisons and Analyses

Large language models (LLMs) have demonstrated strong performance across a broad spectrum of tasks, including code generation (Sadik & Govind, 2025; Wang et al., 2025), question answering (Bottou et al., 2018; Bai et al., 2024; Sun et al., 2024), and mathematical reasoning (Minaee et al., 2024; Wang et al., 2026; Muennighoff et al., 2025; Sun et al., 2025). Nevertheless, achieving peak performance in specialized domains often still requires targeted post-training adaptation (Tie et al., 2025; Sun et al., 2026; 2021). Supervised fine-tuning and instruction tuning (Raffel et al., 2020; Devlin et al., 2019; Zang et al., 2025; Sanh et al., 2022; Chung et al., 2024; Ouyang et al., 2022) can effectively align model behavior to task objectives using labeled data or instructional demonstrations, but they may exhibit exposure bias and limited robustness under distribution shifts or novel reasoning contexts (Touvron et al., 2023; Ballon et al., 2025). Reinforcement learning (RL) has emerged as a central paradigm for post-training, enabling LLMs to optimize directly for task-level rewards and human preference signals (Ouyang et al., 2022; Bai et al., 2022). Within this family, Group Relative Policy Optimization (GRPO) (Shao et al., 2024) has gained considerable attention for its scalability: it performs policy updates using relative advantages computed within groups of sampled candidate responses, reducing computational overhead while supporting diverse reward formulations, including preference and process-level signals. Building on this framework, a number of GRPO-style variants further improve efficiency and plasticity through refined rewards and regularization strategies. For example, Dr.GRPO (Liu et al., 2025) improves token efficiency while preserving reasoning performance; GVPO (Zhang et al., 2026) incorporates an analytic view of KL-constrained reward maximization into gradient weighting; L2T (Wang et al., 2026) proposes an information-theoretic RL fine-tuning approach to achieve stronger reasoning with fewer tokens; and GCPO (Gu et al., 2026) aligns the policy to the causal reference distribution through reward correction, achieving great performance on multiple benchmarks. Despite this progress, existing methods typically treat the policy-improvement objective and the KL-based behavior-preservation constraint as being reconciled by deterministic gradient addition (Ziems et al., 2025; Ge et al., 2025). In high-dimensional parameter spaces, however, these two forces can conflict within specific subspaces: the clipped-surrogate term encourages policy shifts toward higher-reward responses, whereas the KL regularizer pulls the policy back toward a reference model. Such local antagonism may lead to unstable optimization trajectories and ultimately suboptimal solutions. To address this issue, we propose Probabilistic Conflict Resolution (PCR). It re-evaluates the update direction as a Bayesian fusion of uncertain gradient distributions, rather than a deterministic collision between the improvement and preservation gradients, thus improving the reasoning performance.

Furthermore, while the mathematical framework of PCR naturally extends to other gradient conflict scenarios such as standard Multi-Task Learning (MTL), our primary contribution lies in identifying and formulating this specific problem

within the LLM post-training paradigm. In standard MTL, gradient conflicts are widely recognized because distinct task labels inherently compete. However, their manifestation in LLM post-training is much less transparent. We highlight that the destructive geometric interference between the RL learning signal (reward plasticity) and the regularization signal (KL penalty stability) during GRPO is a subtle yet critical bottleneck. By formalizing this LLM-specific challenge, we arrive at a probabilistic framework whose algorithmic choices are carefully tailored to the practical realities of LLMs. Unlike generic MTL optimizers, PCR specifically leverages the high-variance stochasticity of GRPO group samples. Moreover, it is designed to operate with minimal memory overhead and is selectively applied to MLP layers to align with the mechanistic understanding of knowledge storage in LLMs. Thus, our work bridges universal geometric optimization principles with the specific, critical challenges of LLM alignment.

# I. Broader Impacts and Limitations

## I.1. Broader Impacts

In this work, we identified one root cause of training instability in Group Relative Policy Optimization (GRPO): the local antagonism between advantage-driven policy gradients and KL-based constraints, which leads to destructive gradient interference in high-dimensional subspaces. To resolve this, we introduced Probabilistic Conflict Resolution (PCR), a novel Bayesian framework that re-conceptualizes gradient updates as the fusion of uncertain probabilistic signals rather than deterministic Euclidean addition. Our extensive evaluations demonstrate the effectiveness of PCR. By effectively reconciling the trade-off between exploration and reference maintenance, PCR paves the way for more robust and efficient post-training of LLMs. It also provides an effective way to improve stability.

## I.2. Limitations

While PCR demonstrates robust effectiveness and generalization across models ranging from 1.5B to 7B parameters, our empirical evaluation is currently constrained to this parameter scale due to computational resource limitations. The optimization dynamics of ultra-large-scale models (e.g., 70B or 405B parameters) may differ from those of smaller models. It remains an open question whether the marginal utility of our mechanism diminishes as the model capacity increases, also our focus in the future.

# J. Implementation Details

Our implementation leverages a modified version of the Verl library as the core codebase. For large-scale model optimization, we incorporated DeepSpeed 0.13.1 with ZeRO-3 stage optimization. Furthermore, to accelerate training throughput, we enabled vLLM acceleration, capping GPU memory utilization at 80%. Entire training pipelines were conducted using BF16 mixed-precision. We initialized our experiments by loading pre-trained base weights directly from Hugging Face for the following architectures, including Qwen2-7B-Instruct, DeepSeek-R1-Distill-Qwen-1.5B, DeepSeek-R1-Distill-Qwen-7B, and DeepScaleR-1.5B-Preview, etc. To address specific mathematical reasoning tasks, distinct fine-tuning regimens were applied to select models. The DeepSeek-R1-Distill-Qwen-1.5B variant underwent fine-tuning using a 4,000-sample random subset from the NuminaMath dataset. The DeepScaleR-1.5B-Preview model was subjected to a rigorous two-stage fine-tuning process, i.e., first trained on a general dataset of 40,000 math problem-solution pairs, followed by a second phase utilizing 919 problems sourced from AIME benchmarks. Across experiments, we maintained a standardized training configuration. The learning rate was set to $1e \times 10^{-6}$, regulated by a cosine scheduler featuring a 0.1 warm-up ratio. The effective batch size was maintained at 256. We established strict token limitations, defining a maximum prompt length of 4,096 tokens and an overall maximum generation budget of 16,384 tokens. The hyper-parameters $\eta$ is set to $5 \times 10^{-5}$ selected by experiments. All experiments are run on A100 GPU clusters.

# K. Benchmark Datasets

To rigorously evaluate our models across different facets of complex reasoning, following (Gu et al., 2026), we utilize a diverse suite of benchmarks. These datasets can be broadly classified into two primary domains: mathematical deduction and algorithmic code generation. The mathematical evaluation relies on four distinct datasets, including AIME24-25, AMC, MATH500 (Hendrycks et al., 2021), and MinervaMATH (Lewkowycz et al., 2022), ranging from standard curriculum problems to high-level competitions. Parallel to this, we assess programming capabilities using the HumanEval benchmark (Chen et al., 2021). Below, we detail the characteristics and composition of each selected benchmark.

*Table 2.* Pass@1 performance on various math reasoning benchmarks. We compare base models trained with different fine-tuning approaches. The best results are highlighted in **bold**. Note that our method can be plug-and-play, highlighted in blue.

| Base model + Method | AIME 2024 | AIME 2025 | AMC 2023 | MATH500 | MinervaMATH | Avg. |
|---|---|---|---|---|---|---|
| **DeepScaleR-1.5B-Preview** | 42.8 | 36.7 | 83.0 | 85.2 | 24.6 | 54.5 |
| GVPO (Zhang et al., 2026) | 46.1 (+3.3) | 39.7 (+3.0) | 83.6 (+0.6) | 85.7 (+0.5) | 25.3 (+0.7) | 56.1 (+1.6) |
| MRT (Qu et al., 2025) | 47.2 (+4.4) | 39.7 (+3.0) | 83.1 (+0.1) | 85.1 (-0.1) | 24.2 (-0.4) | 55.9 (+1.4) |
| GCPO (Gu et al., 2026) | 46.7 (+3.9) | 40.3 (+3.6) | 84.1 (+1.1) | 86.3 (+1.1) | 25.9 (+1.4) | 56.8 (+2.3) |
| GVPO+PCR | 48.5 (+5.7) | 41.8 (+5.1) | 85.6 (+2.6) | 87.6 (+2.4) | 27.7 (+3.1) | 58.2 (+3.7) |
| MRT+PCR | **49.5 (+6.7)** | 42.3 (+5.6) | 85.2 (+2.2) | 86.8 (+1.6) | 26.5 (+1.9) | 58.1 (+3.6) |
| GCPO+PCR | 49.1 (+6.3) | **43.0 (+6.3)** | **86.3 (+3.3)** | **88.4 (+3.2)** | **28.4 (+3.8)** | **59.0 (+4.5)** |
| **DeepSeek-R1-Distill-Qwen-1.5B** | 28.7 | 26.0 | 69.9 | 80.1 | 19.8 | 44.9 |
| GVPO (Zhang et al., 2026) | 30.6 (+1.9) | 28.2 (+2.2) | 71.5 (+1.6) | 80.5 (+0.4) | 23.1 (+3.3) | 46.7 (+1.8) |
| MRT (Qu et al., 2025) | 30.3 (+1.6) | 29.3 (+3.3) | 72.9 (+3.0) | 80.4 (+0.3) | 22.5 (+2.7) | 47.1 (+2.2) |
| GCPO (Gu et al., 2026) | 31.0 (+2.3) | 29.0 (+3.0) | 71.8 (+1.9) | 81.6 (+1.5) | 23.4 (+3.6) | 47.4 (+2.5) |
| GVPO+PCR | 32.5 (+3.8) | 30.8 (+4.8) | 73.5 (+3.6) | 81.9 (+1.8) | 25.6 (+5.8) | 48.9 (+4.0) |
| MRT+PCR | 32.2 (+3.5) | 31.5 (+5.5) | 75.2 (+5.3) | 81.8 (+1.7) | 24.8 (+5.0) | 49.1 (+4.2) |
| GCPO+PCR | **33.1 (+4.4)** | **31.6 (+5.6)** | 74.5 (+4.6) | **82.9 (+2.8)** | **25.8 (+6.0)** | **49.6 (+4.7)** |
| **DeepSeek-R1-Distill-Qwen-7B** | 55.5 | 50.2 | 85.1 | 87.4 | 42.1 | 64.1 |
| GVPO (Zhang et al., 2026) | 57.5 (+2.0) | 52.1 (+1.9) | 86.3 (+1.2) | 88.5 (+1.1) | 44.2 (+2.1) | 65.7 (+1.6) |
| MRT (Qu et al., 2025) | 57.0 (+1.5) | 52.4 (+2.2) | 86.0 (+0.9) | 88.4 (+1.0) | 44.3 (+2.2) | 65.6 (+1.5) |
| GCPO (Gu et al., 2026) | 58.3 (+2.8) | 53.0 (+2.8) | 87.3 (+2.2) | 89.1 (+1.7) | 45.0 (+2.9) | 66.5 (+2.4) |
| GVPO+PCR | 59.8 (+4.3) | 54.5 (+4.3) | 88.2 (+3.1) | 90.1 (+2.7) | 47.2 (+5.1) | 68.0 (+3.9) |
| MRT+PCR | 58.9 (+3.4) | 54.8 (+4.6) | 87.4 (+2.3) | 90.5 (+3.1) | 47.5 (+5.4) | 67.7 (+3.6) |
| GCPO+PCR | **60.5 (+5.0)** | **55.2 (+5.0)** | **89.1 (+4.0)** | **90.9 (+3.5)** | **47.8 (+5.7)** | **68.7 (+4.6)** |

*AIME24-25* comprises a focused collection of 30 numerical-answer problems sourced from the 2024 and 2025 iterations of the American Invitational Mathematics Examinations (15 questions per year). Representing a significant step up in complexity relative to the AMC, these tasks demand advanced problem-solving skills across core domains such as number theory, combinatorics, geometry, and algebra.

*AMC* aggregates a comprehensive set of 975 multiple-choice problems spanning 39 distinct competitions from the AMC10 and AMC12 series, designed for 10th and 12th-grade students, respectively. This dataset offers a broad spectrum of difficulty, covering foundational topics from basic algebra and geometry to introductory concepts in probability and combinatorics, serving as a robust baseline for evaluating mathematical logic.

*MATH500* serves as a representative evaluation set consisting of 500 problems randomly sampled from the extensive MATH benchmark. It covers seven distinct mathematical disciplines, ranging from prealgebra and number theory to intermediate algebra and precalculus. Crucially, each entry is annotated with a difficulty level and a full step-by-step solution, facilitating granular performance analysis across different complexity tiers.

*MinervaMATH* features a large-scale corpus of 12,500 high-school level problems modeled after mathematics competitions. Spanning a complete curriculum from prealgebra through precalculus, each question is augmented with detailed solution steps, supporting the evaluation of the model's ability to generate coherent long-form derivations.

*HumanEval* is utilized to assess algorithmic coding capabilities, consisting of 164 unique Python programming tasks. For each instance, the model is provided with a function signature and a natural language docstring, with the objective of synthesizing a strictly functional implementation. The performance is quantified using the $Pass@k$ metric, which evaluates the success rate of the generated code against a suite of unit tests.

## L. Additional Experiments and Discussion

### L.1. More Results of Performance Analyses

We further investigate the transferability of PCR to determine whether it can serve as a universal plug-and-play module across different post-training frameworks and base models. To this end, we apply PCR on top of different post-training architectures, i.e., GVPO, MRT, and GCPO. As reported in Table 2, PCR demonstrates consistent and significant performance gains across all settings, validating its robustness regardless of the underlying model capability or pre-training focus. Specifically, (1)

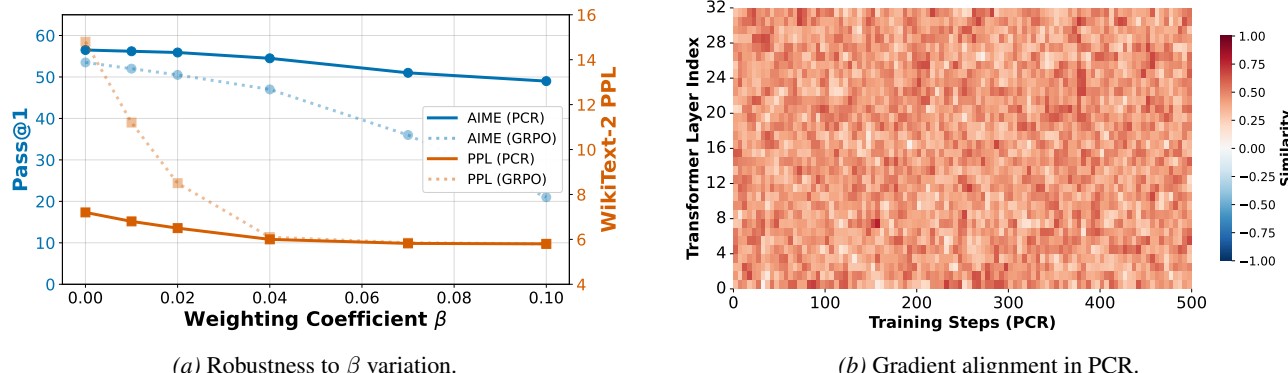

*(a)* Robustness to $\beta$ variation.                    *(b)* Gradient alignment in PCR.

*Figure 6.* Effectiveness of PCR. (a) illustrates the comparison of AIME accuracy and WikiText-2 perplexity between PCR and GRPO under varying weighting coefficients $\beta$. (b) visualizes the layer-wise similarity between the generation gradients and stability gradients throughout the PCR training process.

*Table 3.* Ablation study on the core components of PCR. We compare PCR against baselines that selectively remove (a) uncertainty estimation, (b) Bayesian formulation, and (c) the soft projection rule.

| Method | Components Active | AIME 2024 (Pass@1) | MATH500 (Pass@1) |
|---|---|---|---|
| Standard GRPO | None | 44.5 | 84.9 |
| 1. Fixed Soft Proj | (c) only | 46.8 | 85.3 |
| 2. Uncertainty + Hard Proj | (a) only | 46.5 | 85.6 |
| 3. Learnable Soft Proj | (a) + (c) | 47.5 | 85.8 |
| PCR (Ours) | (a) + (b) + (c) | **48.1** | **87.0** |

on the DeepScaleR-1.5B baseline, while competitive methods like GVPO and GCPO achieve average improvements of 1.6% and 2.3% respectively, PCR pushes the boundary further with a 3.2% increase, reaching a state-of-the-art average accuracy of 57.7%; (2) on the smaller DeepSeek-R1-Distill-Qwen-1.5B, PCR delivers the most substantial relative gain of 3.6%, outperforming the second-best method GCPO by a clear margin of 1.1%; and (3) scaling to the 7B parameter size, our method maintains its superiority, boosting the average performance to 67.6% compared to 66.5% achieved by GCPO.

Crucially, the experimental results highlight that PCR effectively addresses the limitations of existing group-relative optimization methods. Standard approaches such as GRPO and MRT often struggle to balance stability and exploration, occasionally leading to performance degradation on specific datasets (e.g., GRPO drops 1.5% on AMC 2023). In contrast, PCR achieves positive gains across every single dataset-model combination, indicating a more stable optimization trajectory. This consistency is particularly evident in high-difficulty reasoning tasks. For instance, on the challenging AIME 2024 and MinervaMATH benchmarks, PCR outperforms all baselines by approximately 1%-2% across different model scales. These findings suggest that the constraint-aware mechanism in PCR does not merely overfit to specific distributions but enhances the fundamental reasoning capability of the policy.

To further verify the universality of our approach, we extended the evaluation to code generation tasks, which share the rigorous logical structure and verifiable constraints characteristic of mathematical reasoning. We assessed the performance of PCR on standard code benchmarks (e.g., HumanEval) using the same plug-and-play integration strategy across different baselines. As illustrated in Figure 7, PCR consistently boosts the Pass@1 accuracy of established post-training methods, including GVPO, MRT, and GCPO, in both 0-shot and 5-shot settings. For instance, even with strong baselines, applying PCR yields a distinct performance margin, pushing the success rate significantly higher (e.g., surpassing 70% in 5-shot scenarios for GCPO). These results demonstrate that the constraint-aware mechanism of PCR is not limited to the math domain; rather, it effectively generalizes to programming tasks, serving as a robust, domain-agnostic enhancement for improving the executability and correctness of code generation models.

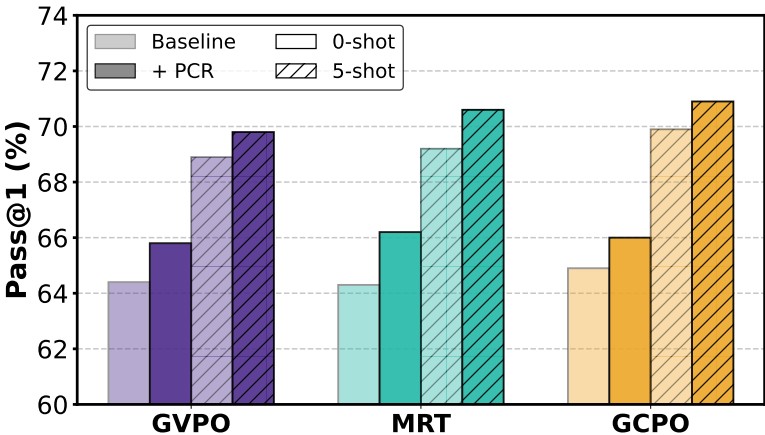

*Figure 7.* The plug-and-play of PCR on code reasoning tasks. We record the 1-shot and 5-shot results on HumanEval.

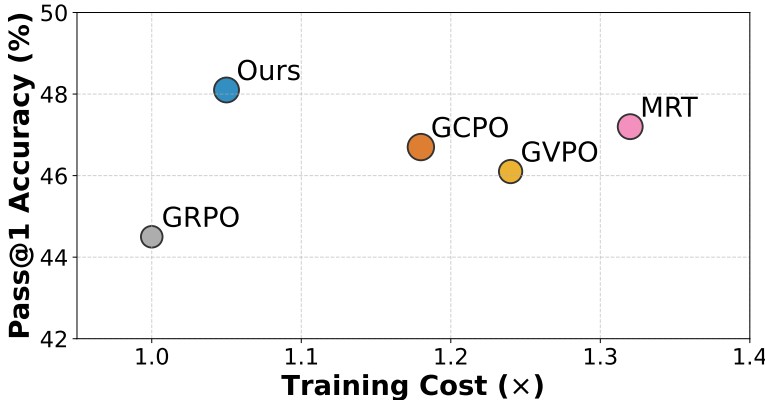

*Figure 8.* The trade-off performance of different post-training methods.

## L.2. More details of Visualization Analysis

To demonstrate that our proposed PCR framework effectively mitigates the optimization pathologies identified in Section 3.3, we repeated the diagnostic experiments on the same DeepSeek-R1-Distill-Llama-8B backbone. The results indicate that PCR successfully decouples the antagonistic objectives, transforming the optimization landscape from a destructive interference regime into a cooperative alignment regime.

First, we examine the sensitivity of model performance to the hyperparameter $\beta$. As illustrated in Figure 6(a), unlike GRPO, which suffers from a sharp collapse in linguistic capability (PPL explosion) at low $\beta$ values, PCR maintains a robust linguistic baseline even when the constraint is relaxed. Crucially, PCR achieves a superior Pareto improvement: at $\beta = 0.04$, it yields a significantly higher Pass@1 accuracy compared to GRPO, while keeping the WikiText-2 PPL nearly constant. This flattened sensitivity curve suggests that PCR provides a safe optimization corridor, eliminating the need for exhaustive hyperparameter grid search and reducing the risk of catastrophic forgetting.

Second, to pinpoint the physical source of this stability, we visualize the cosine similarity between the plasticity and stability gradients under PCR training. The heatmap in Figure 6(b) presents a stark contrast to the conflict-ridden landscape of standard GRPO (refer back to Figure 1(b)). Under PCR, the previously antagonistic middle-to-deep MLP layers now exhibit consistently positive cosine similarity. This indicates that the PCR update rule effectively projects the reasoning gradient onto a subspace that is congruent with the general knowledge manifold. By converting gradient conflict into gradient synergy, PCR ensures that improvements in reasoning logic reinforce, rather than cannibalize, the model's linguistic foundations.

*Table 4.* Sensitivity to group size on AIME 2024.

| Method | $G = 16$ | $G = 8$ | $G = 4$ |
|---|---|---|---|
| GRPO | 44.5 | 38.2 | 25.4 |
| PCR (Ours) | **48.1** | **46.5** | **43.2** |

*Table 5.* Sensitivity to reward variance on AIME 2024.

| Method | 0% Noise | 10% Noise | 20% Noise |
|---|---|---|---|
| GRPO | 44.5 | 36.4 | 22.8 |
| PCR (Ours) | **48.1** | **45.6** | **42.1** |

*Table 6.* Performance comparison of PCR and PCGrad in MTL settings.

| Method | MultiMNIST (Task L Acc) | MultiMNIST (Task R Acc) | CelebA (Avg. Error) |
|---|---|---|---|
| Single Task Baseline | 88.2% | 87.5% | 8.92% |
| PCGrad (Deterministic) | 89.6% | 88.8% | 8.15% |
| PCR (Probabilistic) | **90.4%** | **90.9%** | **7.59%** |

## L.3. Computational Efficiency and Complexity

To assess the practical feasibility of probabilistic conflict resolution in large-scale post-training, we conduct a comprehensive analysis of computational cost versus model performance. We evaluate the training efficiency by measuring the total wall-clock time required for convergence under identical hardware configurations (e.g., same GPU cluster, batch size, and precision). To facilitate a direct comparison, we normalize the training cost of the standard GRPO baseline to $1.0\times$. The relationship between computational overhead (x-axis) and reasoning accuracy (Pass@1, y-axis) is visualized in Figure 8. We can observe that PCR effectively breaks the trade-off usually observed in prior works, i.e., achieves the highest pass@1 performance with lower cost. It offers a high-return solution for post-training.

## L.4. Scalability to Larger Update Magnitudes

To further explore the performance of our method, we investigate the behavior of PCR under varying update magnitudes by scaling the learning rate $\eta$. As illustrated in Figure 4c, PCR demonstrates remarkable stability and a consistent upward trend as the update scale increases. Notably, even when the learning rate is scaled by $1.5\times$ or $2.0\times$, the model continues to achieve performance gains, reaching a peak score of 48.5%. This behavior suggests that the probabilistic arbitration of PCR effectively filters out the detrimental noise and conflicting signals that typically plague aggressive optimization, allowing the model to safely traverse the parameter space with larger steps to capture more complex reasoning patterns.

## L.5. Detailed Ablation on PCR Components

To definitively isolate the benefits of (a) uncertainty estimation, (b) the Bayesian formulation, and (c) the soft projection rule, we design a comprehensive ablation study on the DeepScaleR-1.5B-Preview backbone. We compare PCR against baselines that selectively remove these core components. The results are provided in Table 3. We can observe that baseline 1 tests soft projection without uncertainty estimation. It improves over GRPO but remains clearly worse than PCR, since gradient signal-to-noise varies over time and across layers. As Figure 5(a) shows, different layers require different projection strengths, which a fixed soft rule cannot adapt to. Baseline 2 pairs uncertainty estimation with hard projection, but still fails because it completely removes the conflicting component once triggered. Under noisy constraints, this is overly rigid. Soft projection is necessary to retain partial exploration signals while reducing harmful conflict. Baseline 3 uses a learnable uncertainty-aware soft projection instead of our closed-form Bayesian PCR. Despite its extra flexibility, it still underperforms PCR. This supports Theorem 5.1: PCR is not heuristic, but the MMSE-optimal solution for balancing constraint bias and exploration variance. In conclusion, the empirical evidence clearly validates that it is the mathematically optimal synthesis of (a), (b), and (c) that unlocks the superior and stable reasoning gains.

## L.6. Sensitivity to Group Size and Reward Variance

The quality of the uncertainty estimates, group size, and reward variance naturally lead to high gradient variance. When the precision of $\lambda_{pla}$ drops due to high noise, $k$ automatically decays toward 0. Therefore, under worst-case scenarios, PCR safely suppresses unreliable updates and gracefully defaults to protecting pre-trained knowledge. We utilize a computationally cheap variance approximation, where PCR achieves robust arbitration simply by capturing the relative signal-to-noise ratio, without relying on complex estimation techniques. We evaluate DeepScaleR-1.5B-Preview across

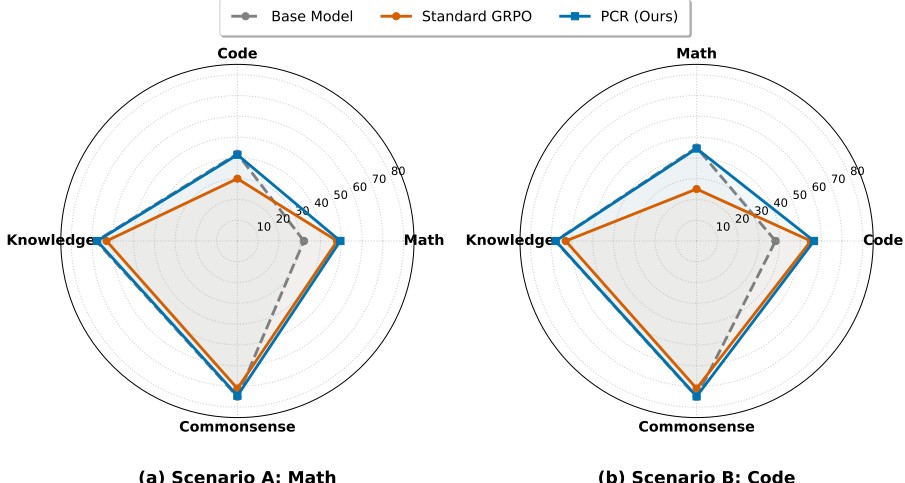

*Figure 9.* Impact on general capabilities. We evaluate the model's cross-domain performance after post-training on specific tasks using radar charts. The gray area represents the pre-trained base model.

varying group sizes and reward noise on the AIME 2024 dataset. Since group size affects uncertainty estimation, we report results only for different group sizes. The results in Table 4 and Table 5 demonstrate its advantages.

### L.7. Generalization to Multi-Task Learning

Geometric conflict resolution has been extensively explored in standard deep learning, most notably in Multi-Task Learning (MTL). The methods like PCGrad resolve conflicts by projecting gradients geometrically to prevent destructive interference. To definitively prove that PCR's underlying Bayesian conflict-resolution mechanism generalizes far beyond GRPO, we directly applied it to standard MTL settings. We adapt PCR by treating the distinct task losses as the conflicting objectives. Since MTL lacks group samples, we estimate the gradient variance across the mini-batch (i.e., per-sample gradient variance) to capture the stochasticity of the update direction. We evaluate this probabilistic formulation against PCGrad (Yu et al., 2020) on two widely adopted MTL benchmarks, i.e., MultiMNIST and CelebA. As shown in Table 6, the empirical results confirm our theoretical hypothesis. Even in standard supervised learning regimes, PCGrad is strictly suboptimal when mini-batch gradients exhibit high variance or noise, whereas our probabilistic projection safely and effectively arbitrates gradient conflicts.

### L.8. Impact on General Capabilities

A critical challenge in post-training via reinforcement learning is the risk of catastrophic forgetting, where the model overfits to the target reward signal (plasticity) at the expense of its pre-trained general capabilities (stability). To evaluate whether PCR can mitigate this issue, we conduct cross-domain evaluations under two distinct post-training scenarios.

Specifically, we follow the experimental settings in Section J, utilizing the pre-trained base model and subject it to post-training in two separate domains: (1) Scenario A (i.e., math-targeted): The model is trained on mathematical reasoning using AIME. We then evaluate its performance on the target task, and three unseen downstream tasks: Coding (i.e., using HumanEval (Chen et al., 2021)), General Knowledge (i.e., using MMLU (Hendrycks et al., 2020)), and Commonsense Reasoning (i.e., using HellaSwag). (2) Scenario B (i.e., code-targeted): The model is trained on code generation (i.e., using MBPP). We evaluate it on the test data and three unseen tasks: Math500, MMLU, and HellaSwag. The results are visualized in the radar charts of Figure 9. In Scenario A (as shown in Figure 9(a)), while standard GRPO achieves gains in math reasoning tasks, it suffers from a significant degradation in coding (-12%) and MMLU (-5%), indicating severe forgetting. In contrast, PCR matches or exceeds GRPO's performance on the target math task while maintaining the coding and MMLU scores at the level of the pre-trained baseline. A similar pattern is observed in Scenario B (as shown in Figure 9(b)), where GRPO improves code generation but causes a collapse in Mathematical reasoning capabilities. PCR successfully decouples the learning of new skills from the overwriting of existing knowledge. These findings confirm that PCR effectively resolves the gradient conflicts between the reward objective and the pre-trained data. By dynamically arbitrating updates, PCR enables the model to improve specific reasoning capabilities without sacrificing its foundational general intelligence.

