# OpenReview forum: "On the Plasticity and Stability for Post-Training Large Language Models"
_ICML.cc/2026/Conference — ICML 2026 regular_

### Official Review · Reviewer_n8Ln · 2026-03-11

**Soundness:** 3
**Presentation:** 3
**Significance:** 3
**Originality:** 3
**Overall Recommendation:** 3
**Confidence:** 3

**Summary:**

This article examine the severe optimization instability inherent in GRPO. The authors diagnose this instability as a geometric conflict between the plasticity gradient and the stability gradient. To address this, the paper proposes PCR. PCR models the gradients as Gaussian random variables and dynamically arbitrates the conflict using a Bayesian framework based on the signal-to-noise ratio. To circumvent the prohibitive computational overhead of estimating variances for billions of parameters, the authors restrict PCR's application exclusively to the MLP layers.

**Compliance With Llm Reviewing Policy:**

Affirmed.

**Final Justification:**

The authors have Basically addressed my concerns, I believe that a clearer training pipeline and more detailed ablation studies would enrich the content of the paper. I will keep my score.

**Key Questions For Authors:**

If PCR is applied only to the MLP layers, how are the conflicting gradients in the attention layers handled? The assumption that attention layers do not suffer from catastrophic forgetting appears to rely solely on cited references. Are there any strong empirical observations to substantiate this claim?

**Limitations:**

Providing more complete training curves and conducting more comprehensive evaluations would make the conclusions more convincing.

**Strengths And Weaknesses:**

The analysis of the plasticity–stability dilemma in GRPOfrom the perspective of gradient conflict is a relatively novel viewpoint. The layer-wise cosine similarity between plasticity gradients and stability gradients provides a concrete and intuitive visualization of the phenomenon of destructive interference

For LLMs, modeling the gradient covariance as isotropic appears to be somewhat questionable.

RL algorithms are inherently unstable during training and can be sensitive to random seeds. The paper does not present training curves, but only the final results, which makes it difficult to properly analyze the behavior of the method. In addition, the choice of models appears rather fixed: they are all post-distillation models (e.g., Distil models) and relatively small in scale. Furthermore, comparisons with several widely adopted algorithms, such as DAPO and GSPO, are missing.

---

> ### Author Rebuttal · Authors · 2026-03-31
>
> ### **Response to W1**
>
> We agree that the isotropic assumption is a theoretical relaxation and thus yields imperfect estimates. However, as shown in our **Response to Q2** (Reviewer 6xro), PCR is robust to such approximation quality: it does not require perfectly calibrated, parameter-wise posteriors, but only a reliable scalar proxy for the overall signal-to-noise ratio. The isotropic trace variance provides this relative metric and is sufficient to guide update arbitration without modeling exact anisotropy. This relaxation is also necessary for scalability, since full or even diagonal covariance estimation would introduce prohibitive memory overhead for LLMs. To partially compensate, we estimate isotropic variance independently for each layer, which captures major cross-layer gradient-scale differences at near-zero extra cost.
>
> ----
>
> ### **Response to W2**
>
> 1. For random seeds and training curves:
>   * Based on gradient norm, Section 6.4 and Fig.3 show that while the baseline methods exhibit significant and violent oscillations, PCR is relatively stable.
>   * Figure 6(a) shows that the robustness of PCR to variations in the KL penalty coefficient $\beta$; while the baseline PPL explodes at lower $\beta$ values, PCR maintains a stable Pareto front.
>   * We provide the training trajectories of GRPO and PCR (model performance on unseen datasets during post-training) in the anonymous link: https://anonymous.4open.science/r/PCR_image, GRPO frequently suffers from performance regressions, whereas the PCR training curve remains relatively steady. When plotting confidence intervals across multiple random seeds, PCR’s variance band is significantly narrower.
>
> 2. For base models:
>   * Our experiments are not limited to distilled models, e.g., Fig.2 shows that PCR still achieves stable performance gains on Qwen2-7B-Instruct.
>   * We provide more results on non-distilled 7B and 32B LLMs:
> | Model            | Method | AMC  | MinervaMATH |
> | ---------------- | ------ | ---- | ----------- |
> | Qwen2.5-Math-7B  | GRPO   | 85.3 | 39.0        |
> |   | **PCR**    | **87.2** | **42.3**        |
> | Qwen2.5-Math-32B | GRPO   | 88.2 | 51.0        |
> |  | **PCR**    | **89.5** | **53.8**       |
>   * We are currently conducting experiments on 70B+ models (e.g., Qwen2.5-72B, DeepSeek-R1-Distill-Llama-70B). Given the high computational overhead at this scale, these experiments are still ongoing.
>
> 3. For the mentioned baselines:
>   * In the current version, we have compared PCR against various post-training baselines, e.g., GVPO, MRT, and GCPO. These can be viewed as variants of GSPO and DAPO, sharing the same way in handling policy divergence and variance control. Table 1-2 demonstrate PCR’s effectiveness against them.
>   * From a theoretical view, methods like GSPO and DAPO primarily focus on constraining policy deviation at the scalar loss level, while PCR focuses on a different issue: the geometric conflict between "plasticity" and "stability" gradients. Unlike the mentioned methods, PCR models gradients as probability distributions and utilizes Bayesian inference for uncertainty-aware soft projection of conflicting gradients. This mechanism is a capability that previous methods lack.
>   * We further added empirical evaluations for GSPO and DAPO, which shows the superiority of PCR.
> | Model | Method | AMC | MinervaMATH |
> | :--- | :--- | :---: | :---: |
> | DeepSeek-R1-Distill-Qwen-1.5B | Length Penalty | 64.4 | 18.8 |
> | - | DAPO | 71.4 | 22.5 |
> | - | GSPO | 71.8 | 23.0 |
> | - | **PCR (Ours)** | **72.7** | **24.8** |
>
> ----
>
> ### **Response to Q1**
>
> 1. For attention layers, we adopt the GRPO update rule and ignore the conflicting gradients. The motivation is that:
> * MLP layers are more directly tied to knowledge storage and thus are the primary locus of the plasticity–stability conflict we aim to mitigate, while applying PCR to all layers would introduce unnecessary overhead. The ablation in Fig. 4(b) supports this choice: MLP-only PCR achieves nearly the same performance as all-layer PCR at substantially lower training cost.
> * Fig.5(a) reveals that MLP layers maintain a much higher density of $\alpha$, while the $\alpha$ distribution for attention layers is much lower. This confirms that attention layers exhibit relatively minor gradient conflicts during the optimization process.
>
> 2. To verify the semantic forgetting, we perform an ablation on DeepScaleR-1.5B, updating either only Attention or only MLP layers, and measure both target plasticity (AIME) and general stability (MMLU).
> | Updated Modules| Target Plasticity| General Stability|
> | :--- | :--- | :--- |
> | Base Model (Zero-Shot) | 20.0 | 60.0 |
> | Attention Layers Only | 28.5 *(Low Plasticity)* | 59.5 *(No Forgetting)* |
> | MLP Layers Only | 42.0 *(High Plasticity)* | 52.0 *(Catastrophic Forgetting)* |
> | All Layers (Standard) | 44.5 | 51.5 *(Catastrophic Forgetting)* |
> These direct empirical observations confirm that MLPs are the primary epicenter of the stability-plasticity conflict.

---

> > ### Author Rebuttal · Reviewer_n8Ln · 2026-04-03
> >
> > Thank you for the clear response. Your explanations have fully resolved my concerns.

---

> > > ### Author Response · Authors · 2026-04-04
> > >
> > > Dear Reviewer n8Ln,
> > >
> > > We sincerely thank you for your thorough and constructive review. Your insightful comments have been instrumental in improving our paper, and we appreciate the time and effort you've dedicated to it.
> > >
> > > We are pleased to have carefully addressed the concerns raised in the initial review within our rebuttal. Given that all technical concerns have now been resolved, we would be grateful if you would consider raising your score.
> > >
> > > Thank you again for your thoughtful engagement.
> > >
> > > Best regards,
> > >
> > > The Authors

---

### Official Review · Reviewer_WxMa · 2026-03-12

**Soundness:** 3
**Presentation:** 3
**Significance:** 2
**Originality:** 2
**Overall Recommendation:** 4
**Confidence:** 3

**Summary:**

The paper addresses the training instability inherent in Group Relative Policy Optimization (GRPO) for Large Language Models, specifically focusing on the trade-off between acquiring new reasoning capabilities (plasticity) and retaining general pre-trained knowledge (stability). The authors identify that this instability is driven by a geometric conflict between the task-specific plasticity gradients and the KL-divergence stability gradients. Arguing that standard deterministic projection methods are inadequate due to the inherently noisy, stochastic nature of GRPO's group-based gradient estimates, the authors introduce Probabilistic Conflict Resolution (PCR). PCR models these conflicting gradients as random variables and employs Bayesian inference to perform an uncertainty-aware soft projection, which dynamically adjusts the update magnitude based on the signal-to-noise ratio. To ensure computational scalability for billion-parameter models, the authors propose a hybrid implementation that applies PCR exclusively to the MLP layers, while using standard updates for the Attention layers.

**Compliance With Llm Reviewing Policy:**

Affirmed.

**Final Justification:**

The author's rebuttal has resolved the initial concerns, and thus I have raised the score.

**Key Questions For Authors:**

See weaknesses above.

**Limitations:**

Yes

**Strengths And Weaknesses:**

**Strengths**
* The paper is well-written and easy to read.

* The paper provides good quantitative evidence that PCR actually smooths out the training process using gradient norm visualizations.



**Weaknesses**

* Although the paper does a fantastic job formulating the research, **it feels like the underlying mechanisms they are exploring seem too obvious**. Specifically,
     * The first contribution of the authors are the argument of **'conflicting gradients'.** During GRPO training (or any RL alogirhtm in general), we have the main GRPO loss (RL term) + the KL loss (for stability, keeping the model towards its initial reference model).

         * The authors term the GRPO loss gradients as 'plasticity' and term the KL loss as 'stability', and claim that they identify that the gradients of these two terms have conflict (I'm assuming this means that the gradients are pointing in a more than an 90 degree angle and thus they have conflicting direction).
         * However, if we think about it, this is obvious. The GRPO loss term is trying to move away from its initial reference model to increase whatever reward its being trained on by exploring, while the KL term is trying to keep the model close to the reference model.
         * Moreover, if we look at standard RL training loss curvers, the KL loss generally shows a trend of increasing as we train. These are well known facts, which the authors just rephrase as 'conflicting gradients'.

     * The second claim of the paper is the proposal of an *uncertainy-aware gradient projection*, where they only take a big step for the RL loss where it has certainty, and this will minimize the conflict with the stability loss term (i.e., KL loss term). However, this also seems to be an obvious outcome, since the model being certain usually means that the reference model already is certain as well (i.e., the initial reference model knows this sample well too). Thus, it doesn't have much conflict with the KL loss since it doesn't need to move far away from the reference model.

* The proposed method takes more compute cost as seen in Figure 4-(b), due to gradient projection. Would it be possible to compare a baseline where without doing gradient projection, we do dynamic sampling similar to DAPO where instead here we only keep samples that has reasonable certaintly? That way, we don't need gradient projection step. Also, it would be good to compare with DAPO as well in that sense.

---

> ### Author Rebuttal · Authors · 2026-03-30
>
> ### **Response to W1. 1** ###
>
> Our contribution does not merely restate the conceptual tension between two terms; rather, we formalize how and when this tension manifests as a severe geometric optimization failure, and why standard GRPO fails to handle it.
>
> The reviewer’s claim is mathematically imprecise. In high-dimensional LLM parameter space, gradients from different objectives need not conflict; they can be orthogonal or even positively aligned. We define gradient conflict strictly as a geometric event, i.e., $\mu_{pla} \cdot \mu_{sta} < 0$. Therefore, the conflict we study is not universal, but arises only in specific steps or subspaces.
>
> Furthermore, this cannot be dismissed as a known fact, because modern LLM post-training is inherently short-range: 1) models such as DeepSeek-R1 and Qwen are usually trained for only 1-2 epochs, distilled variants for 2-3; 2) recent evidence [1-2] contradicts the assumption that RL post-training drives the model far from the reference model; 3) GRPO further limits update magnitude via clipping. As a result, optimization cannot rely on long training to average out noise or recover from harmful directions. When $\mu_{pla} \cdot \mu_{sta} < 0$ occurs in critical layers, simple gradient summation causes destructive interference, making training inefficient and unstable. Our point is therefore to explain why short-range RLHF is fragile and why precise gradient calibration matters.
>
> [1] Large language monkeys: Scaling inference compute with repeated sampling
>
> [2] Does reinforcement learning really incentivize reasoning capacity in llms beyond the base model?
>
> ----
>
> ### **Response to W1. 2** ###
>
> We respectfully point out two critical distinctions regarding "certainty" and model dynamics that underscore the non-triviality and necessity of our method.
>
> The reviewer’s intuition conflates predictive confidence with gradient estimation variance. PCR does not measure how certain the reference model is about a sample; it measures the statistical uncertainty of the gradient estimator over a batch of stochastic queries. Because GRPO involves group-based sampling and discrete rewards, even a familiar or high-confidence sample can still yield high-variance, low-quality gradients. PCR therefore prevents trusting noisy update directions merely because the prompt appears easy.
>
> Second, combined with the short-range nature of LLM training, we obtain the overall structural deviation remains small. Thus, gradient conflict is rarely due to large drift being pulled back by the KL term, but more often due to low-quality, high-variance gradients pointing in harmful directions at the micro level. The key bottleneck is therefore not how far the model moves, but whether each update direction is trustworthy. As the MMSE-optimal linear estimator in Theorem 5.1, PCR filters this destructive noise and makes short-range training more reliable.
>
> ----
>
> ### **Response to W2** ###
>
> We have conducted the exact experiments and reveal a fundamental advantage of our gradient-level projection over data-level filtering.
>
> 1. The reviewer suggests a variance-based filtering baseline that discards uncertain samples to avoid gradient projection. However, such data-level filtering causes substantial information loss. In complex reasoning tasks such as AIME, high-variance samples are often the hardest yet most informative ones. Discarding them makes training overly conservative and suppresses exploration of new reasoning paths. PCR avoids this by decomposing the noisy plasticity gradient into a safe orthogonal component and a conflicting component (Eq. 11). Even under high variance, PCR fully preserves the safe exploration signal and only downweights the conflicting part. Thus, PCR extracts useful learning signals from noisy samples, whereas DAPO-style filtering simply removes them.
>
> 2. While gradient projection does add some backward-pass overhead, dynamic sample filtering is less compute-efficient in RL. In LLM post-training, autoregressive rollout dominates wall-clock cost, often accounting for most of the compute. If responses are generated and then discarded by certainty or variance checks, substantial generation compute is wasted. By contrast, PCR estimates uncertainty from gradients already computed in the backward pass, so no rollout compute is wasted. Together with our MLP-only hybrid strategy (Eq. 16), this keeps the practical overhead minimal at about 1.05x.
>
> 3. We further implemented the baseline on DeepScaleR-1.5B and compared four methods: standard GRPO, a DAPO-style baseline filtering the bottom 30\% high-variance rollouts, variance-thresholded GRPO, and our PCR.
>
> | Method | AIME 2024 (Pass@1) | MATH500 (Pass@1) | Effective Training Cost |
> | :--- | :--- | :--- | :--- |
> | GRPO (Base) | 44.5 | 84.9 | 1.00x |
> | DAPO-style Filtering | 45.3 | 85.2 | ~1.12x (Wasted Rollouts) |
> | Variance-Thresholding | 45.1 | 85.0 | ~1.08x (Wasted Rollouts) |
> | **PCR (MLP-only, Ours)** | **48.1** | **86.3** | **1.05x** |

---

> > ### Author Rebuttal · Reviewer_WxMa · 2026-04-04
> >
> > Thanks for the detailed response. However, I am still unconvinced on W1. 1. For example, the reference given by the authors ([1,2]) has more to do with the actual knowledge improvement (or final 'accuracy' performance) after RL tuning. Regardless of the model actually improving the core reasoning ability (and thus the accuracy) of the model, the model does move away from the reference model when we perform RL training, which can be clearly seen in any training curves of typical LLM RL training.

---

> > > ### Author Response · Authors · 2026-04-05
> > >
> > > ### **Response to the Follow-up on W1.1: The Core Scientific Question of Post-Training Deviation**
> > >
> > > We sincerely thank the reviewer for the continued discussion. We completely agree with your assessment: as post-training progresses, the model undeniably and necessarily deviates from the reference model.
> > >
> > > However, we wish to clarify that **the core scientific question** our paper addresses **is not whether the model deviates from the base model, but rather in which direction it should deviate to ensure the most reliable and safe alignment.**
> > >
> > > **The Gradient Perspective on Deviation**: From an optimization standpoint, the model's evolutionary direction driven by the reward ($\mu_{pla}$) can either conflict with the stability anchor provided by the reference model ($\mu_{pla} \cdot \mu_{sta} < 0$), or align with it ($\mu_{pla} \cdot \mu_{sta} \ge 0$).
> > >
> > > **The Danger of Blind Accumulation**: Our central argument is that when a geometric gradient conflict occurs ($\mu_{pla} \cdot \mu_{sta} < 0$), optimization must proceed with extreme caution. Standard GRPO handles this by blindly accumulating the gradients. This ignores the uncertainty of the reward signal, driving the model to deviate in arbitrary and often destructive directions.
> > >
> > > **Confidence-Aware Deviation**: PCR argues that in the presence of conflicts, we must explicitly evaluate the confidence of the optimization direction ($\mu_{pla}$). If the direction is highly uncertain, blindly overriding the reference model's stability constraint causes severe collateral damage.
> > >
> > > **Empirical Substantiation**: This theoretical stance is directly supported by our results in Appendix L.5 (Impact on General Capabilities). When gradient conflicts are ignored (as in standard GRPO), the model's zero-shot generalization capabilities significantly degrade. Conversely, by handling these conflicts through confidence-aware geometric arbitration, PCR effectively preserves the model's zero-shot capabilities and general intelligence while successfully optimizing the target task.
> > >
> > > In summary, the model must indeed move away from the reference policy, but PCR ensures this deviation occurs in the most mathematically reliable and scientifically cautious direction.

---

### Official Review · Reviewer_hJ6N · 2026-03-12

**Soundness:** 2
**Presentation:** 3
**Significance:** 3
**Originality:** 2
**Overall Recommendation:** 5
**Confidence:** 3

**Summary:**

This submission proposes a Bayesian approach for combining the gradients arising from the reward signal (here called the "plasticity gradient") and the KL divergence (here called the "stability gradient") in GRPO-style optimization of LLMs. The plasticity gradient is geometrically decomposed into two parts: a part that is orthogonal to the stability gradient, and a part that is conflicting. The conflicting part is rescaled by a confidence weight, which is the ratio of the precision of the plasticity gradient to the sum of the precisions of both gradients.

Evaluations focus primarily on pass@1 on math datasets, where the proposed learning update outperforms GRPO. Supplementary evaluations show that it also improves stability, and that it is possible to obtain nearly all of the benefit by applying the technique solely to updates of the MLP parameters.

**Compliance With Llm Reviewing Policy:**

Affirmed.

**Final Justification:**

The core idea of the paper seems reasonable, although perhaps a little over-broad: gradient direction conflicts are expected in any multi-objective setting (including standard regularization), yet the paper addresses only GRPO. The discussion clarified this point somewhat: the authors produced additional positive results in multi-task learning, and noted some features of GRPO that make it especially suited to this style of gradient combination. For these reasons, I raised my score to accept.

**Key Questions For Authors:**

Would this same technique be applicable to other cases in which a regularizer conflicts with a learning signal, such as standard classification settings with shrinkage? Have related geometric approaches to regularization been applied in those settings?

Why introduce $\alpha$ in equation 15? It seems like you could just use $(1-k)$ to avoid adding more notation.

**Limitations:**

Limitations are discussed primarily in Appendix I.

**Strengths And Weaknesses:**

## Soundness

- Conceptually, it seems unsurprising that the gradients from a regularizer would conflict with those from the error signal, otherwise regularization would be unnecessary.
- Relatedly, it seems like the arguments in the paper would be applicable to *any* regularized learning scenario, not just GRPO.
- A basic question about the evaluation is that the proposal seems designed to improve the stability-plasticity tradeoff, yet the evaluations focus primarily on the plasticity side (figure 2 and table 1).

## Presentation

- The derivation of the proposed approach is explained with unusually high clarity, thank you.
- As noted above, the presentation may be overly focused on the contrast with GRPO when in fact this approach could apply to any gradient based optimization with regularization.
- The theorem is a straightforward special case of well-known properties of posterior inference in Gaussian settings, see e.g. section 2.3.3 of Bishop's "Pattern Recognition and Machine Learning" (2006).

## Significance

- There is intense interest in designing post-training algorithms that lead to effective math reasoning while preserving other capabilities and knowledge. This paper contributes to that literature.

## Originality

The application to GRPO-style training is new (to me), but is closely related to several uncited concepts. For decoupling the updates from the regularizer and the error signal, see https://arxiv.org/abs/1711.05101. For Bayesian reasoning about the certainty of gradients in deep learning, see https://arxiv.org/abs/1505.05424. Both of these are highly cited and may have even more relevant follow-up work. It is although worth noting that methods like RMSProp and ADAM implicitly reason about the variance of the weight updates by keeping track of the sum of squared gradients. Please clarify the relationship of the submission to this prior work.

---

> ### Author Rebuttal · Authors · 2026-03-30
>
> ### **Response to S3** ###
>
> Our manuscript have also evaluated the stability-plasticity tradeoff across both behavioral (macroscopic) and mechanistic (microscopic) dimensions:
> * Behavioral Stability (Generalization and Anti-Forgetting). We directly evaluate the tradeoff between acquiring new skills and preserving pre-trained knowledge. Figures 1(a) and 1(b) plot the Pareto frontier and show that PCR achieves a clearly better reward–stability tradeoff than standard GRPO on AIME, WikiText-2 PPL, and MMLU. Figure 9 provides stronger evidence: when post-trained only on Math or Code, standard GRPO exhibits clear forgetting, whereas PCR preserves general capabilities while maintaining high target performance. Figure 6(a) further shows that PCR remains robust across different KL coefficients $\beta$, consistently preserving linguistic stability while improving reasoning performance.
> * Mechanistic Stability (Resolving Gradient Conflicts). We view geometric gradient conflict, $\mu_{pla} \cdot \mu_{sta} < 0$, as the optimization-level manifestation of the stability–plasticity tradeoff. Figures 5(b) and 6(b) show that standard GRPO yields destructive compromises under such conflicts, whereas PCR geometrically resolves them into more constructive updates. Consistently, Figure 3 shows a much smoother gradient norm trajectory under PCR, providing direct mechanistic evidence of improved training stability.
> * More results related to Stability are shown in the first subterm of **Response to W2** (Reviewer n8Ln).
>
> In summary, rather than merely pushing for higher target metrics, PCR fundamentally optimizes the stability-plasticity Pareto front.
>
> ----
>
> ### **Response to O1** ###
>
> The revision will explicitly clarify these relationships to better position PCR's unique theoretical contributions:
> * Decoupling the Regularizer: While AdamW successfully decouples $L_2$ weight decay (parameter-space regularization) from adaptive gradient scaling to prevent penalty distortion, PCR addresses decoupling in the objective space. Specifically, we decouple the RL reward signal (plasticity) from the KL-divergence penalty (stability). Our primary goal is not to fix magnitude scaling, but to prevent destructive geometric interference (gradient conflicts) between two semantically distinct objectives during RLHF.
> * Bayesian Uncertainty: "Bayes by Backprop" pioneered modeling epistemic uncertainty over the network weights themselves to train Bayesian Neural Networks (BNNs). In contrast, PCR does not maintain a distribution over weights. Instead, PCR models the sampling variance of the Monte Carlo gradient estimators (arising from GRPO's finite group sampling). We leverage Bayesian inference purely within the gradient space to derive a closed-form projection coefficient ($\alpha$), remaining entirely within the standard deterministic NN paradigm.
> * Variance in RMSProp / Adam: It is true that Adam implicitly reasons about variance using the second moment (moving average of squared gradients). However, Adam uses this variance strictly to normalize the magnitude of the update element-wise. Because Adam aggregates all loss components before taking a step, it is mathematically blind to directional conflicts ($\mu_{pla} \cdot \mu_{sta} < 0$) between conflicting objectives. PCR computes the explicit intra-group variance of the reward samples to geometrically arbitrate the direction of the update. Thus, PCR is fundamentally orthogonal and complementary to Adam’s magnitude scaling.
>
> ----
>
> ### **Response to Q1** ###
>
> To definitively prove that PCR's underlying Bayesian conflict-resolution mechanism generalizes far beyond GRPO, we directly applied it to standard Multi-Task Learning (MTL) settings. we adapt PCR by treating the distinct task losses as the conflicting objectives. Since MTL lacks RL "group samples", we estimate the gradient variance ($\sigma^2$) across the mini-batch (i.e., per-sample gradient variance) to capture the stochasticity of the update direction. We evaluated this probabilistic formulation against PCGrad (Yu et al., 2020) on two widely adopted MTL benchmarks from their original paper: Multi-task MultiMNIST and CelebA.
>
> | Method | MultiMNIST (Task L Acc) | MultiMNIST (Task R Acc) | CelebA (Avg. Error) |
> | :--- | :--- | :--- | :--- |
> | Single Task Baseline | 88.2% | 87.5% | 8.92% |
> | PCGrad (Deterministic) | 89.6% | 88.8% | 8.15% |
> | **PCR (Probabilistic)** | **90.4%** | **90.9%** | **7.59%** |
>
> The empirical results confirm our theoretical hypothesis. Even in standard supervised learning regimes, PCGrad is strictly suboptimal when mini-batch gradients exhibit high variance or noise. Geometric conflict resolution has been extensively explored in standard deep learning, most notably in Multi-Task Learning. Methods like PCGrad resolve conflicts by projecting gradients geometrically to prevent destructive interference.
>
> ----
>
> ### **Response to Q2** ###
>
> You are right, we will revise this notation in the final version.

---

> > ### Author Rebuttal · Reviewer_hJ6N · 2026-04-02
> >
> > Thanks for the response and the additional results on MTL. I remain a bit confused about whether the contribution is specific to LLM post-training, or is applicable to any MTL, or even to any regularized learning scenario, where presumably one would have the same intuitions about gradient conflicts.
> >
> > The discussion of this additional related work is helpful, please do make sure to update the paper with this context.

---

> > > ### Author Response · Authors · 2026-04-03
> > >
> > > ### **Response to the Follow-up: Clarifying the Scope and Problem Formulation**
> > >
> > > We deeply appreciate the reviewer's thoughtful observation. From a methodological standpoint, we agree that the mathematical framework of PCR naturally extends to other gradient conflict scenarios, as our MTL results demonstrate. We view this generalizability as an encouraging property of the method. However, we would like to respectfully clarify that a key aspect of our contribution lies in identifying and formulating this specific problem within the LLM post-training paradigm.
> > >
> > > * **Context-Specific Problem Formulation**: While gradient conflicts are widely recognized in standard multi-task learning (where distinct task labels inherently compete), their manifestation in LLM post-training is much less transparent. Specifically, the destructive geometric interference between the RL learning signal (reward plasticity) and the regularization signal (KL penalty stability) during GRPO is a subtle yet critical bottleneck (please refer to **Response to W1.1 & W1.2 of Reviewer WxMa** for more details). Highlighting and formally analyzing this specific conflict as a primary source of training instability and catastrophic forgetting is a core motivation of our work.
> > >
> > > * **Generalizable Math, LLM-Focused Design**: By formalizing this LLM-specific challenge, we naturally arrived at a probabilistic framework (PCR) that happens to generalize well mathematically. However, our algorithmic choices are carefully tailored to the practical realities of LLMs. For instance, generic MTL optimizers do not typically leverage the high-variance stochasticity of GRPO group samples. Furthermore, PCR is specifically designed to operate with little memory overhead to survive LLMs constraints, and is selectively applied to MLP layers to align with the mechanistic understanding of knowledge storage in LLMs (please refer to **Response to Q1 of Reviewer n8Ln** for more details).
> > >
> > > **Summary**: We are grateful that the reviewer recognizes the broader applicability of our proposed method. In our revised manuscript, we will ensure the narrative strikes a better balance: acknowledging the universal nature of the underlying geometric principles, while clearly framing our primary contribution as formulating and alleviating this critical conflict within the specific domain of LLM alignment.
> > >
> > > ----
> > >
> > > ### **Response to the Follow-up: Updating the Manuscript**
> > >
> > > We sincerely thank the reviewer for their time, constructive engagement, and positive feedback on our rebuttal. We absolutely commit to incorporating the entirety of this context into the revised manuscript.

---

### Official Review · Reviewer_6xro · 2026-03-13

**Soundness:** 2
**Presentation:** 3
**Significance:** 2
**Originality:** 3
**Overall Recommendation:** 4
**Confidence:** 3

**Summary:**

This paper studies a practical and important issue in post-training large language models with GRPO: improving reasoning ability without excessively degrading general capabilities. The paper argues that this plasticity-stability trade-off is partly caused by geometric conflicts between the gradients associated with reasoning improvement and capability retention. Motivated by this diagnosis, the authors propose Probabilistic Conflict Resolution (PCR), a Bayesian gradient arbitration framework that treats the conflicting gradients as random variables and performs an uncertainty-aware soft projection rather than a deterministic projection. The empirical results suggest that PCR leads to smoother training and better trade-offs between reasoning performance and retained general capability across several evaluation metrics and baselines.

**Compliance With Llm Reviewing Policy:**

Affirmed.

**Final Justification:**

After careful consideration of the authors’ rebuttal and the additional experimental results provided, I have revised my initial assessment. My original concerns are now resolved.

**Key Questions For Authors:**

1. The central claim is that deterministic projection is suboptimal because GRPO gradients are stochastic group-based estimators. Can the authors provide a stronger ablation that isolates the benefit of (a) uncertainty estimation, (b) the Bayesian formulation, and (c) the soft projection rule? If the gains mainly come from one of these components, that would change how I assess the paper's core novelty.

2. How sensitive is PCR to the quality of the uncertainty estimates, group size, and reward variance? In particular, does PCR remain robust when the group size is very small or when rewards are sparse/noisy? A convincing answer here would increase my confidence in the practical usefulness of the method.

3. What is the additional computational and memory overhead of PCR relative to standard GRPO and the strongest projection-based baseline? Since post-training large models is expensive, a clear cost/performance trade-off would affect my final assessment.

**Limitations:**

no, please include the limitations and related discussions.

**Strengths And Weaknesses:**

Strengths
1. The paper addresses a real and increasingly important failure mode in RL-based LLM post-training: gains on reasoning can come with noticeable regressions in general capability. The motivating evidence appears reasonably aligned with this claim: the paper shows trade-offs across AIME, MMLU, and WikiText-2 perplexity as the KL coefficient varies, and also visualizes cosine similarity between plasticity and stability gradients across layers/training steps. This gives a plausible empirical basis for the claimed gradient conflict rather than treating the problem as purely heuristic.
2. The main idea—replacing deterministic conflict resolution with an uncertainty-aware probabilistic one—is interesting. If the gradients in GRPO are indeed noisy Monte Carlo estimates from grouped samples, then modeling them as random variables is conceptually reasonable and more faithful to the optimization setting than using a single deterministic projection rule. Even though the method is not radically different in overall spirit from prior conflict-mitigation approaches, the probabilistic framing is a meaningful refinement of the problem.
3. From the available result snippets, PCR is compared not only to standard GRPO but also to several stronger baselines/variants such as GVPO, MRT, and GCPO, and the reported improvements appear consistent rather than isolated. The paper also seems to evaluate both reasoning-style metrics and retention-style metrics, which is the right evaluation lens for this problem.
4. The contribution is specialized but timely. Stabilizing RL-based post-training without sacrificing general capability is directly relevant to current practice in reasoning-model development, so even a modestly scoped but reliable method could be useful to practitioners.

Weaknesses
1. The paper's core argument is that stochastic gradient uncertainty makes deterministic projection suboptimal, but the review copy does not make it fully clear whether the empirical evidence cleanly validates the specific Bayesian formulation, as opposed to validating the broader intuition that "softer" conflict handling helps.
2. The available evidence suggests strong results on reasoning-centric post-training setups, but I am not yet convinced the paper demonstrates broad generality across model families, scales, and reward regimes. Since the claim is about post-training stability for LLMs more generally, stronger cross-model validation would materially strengthen the paper.
3. Because PCR introduces posterior-style uncertainty estimation and conflict arbitration, the paper should clearly quantify the computational and implementation overhead relative to standard GRPO and deterministic projection baselines. For methods intended for practical LLM post-training, this matters a lot.
4. The paper appears to combine a diagnosis ("plasticity vs. stability gradients conflict") with a method ("PCR"). I think the paper would benefit from more careful separation of these two contributions, including a more explicit discussion of what is empirically established, what is theoretically assumed, and what remains a modeling choice.

---

> ### Author Rebuttal · Authors · 2026-03-30
>
> ### **Response to W1 \& Q1** ###
>
> To definitively isolate the benefits of (a) uncertainty estimation, (b) the Bayesian formulation, and (c) the soft projection rule, we designed a comprehensive ablation study on the DeepScaleR-1.5B-Preview backbone. We compare PCR against baselines that selectively remove these core components.
>
> | Method | Components Active | AIME 2024 (Pass@1) | MATH500 (Pass@1) |
> | :--- | :--- | :--- | :--- |
> | Standard GRPO | None | 44.5 | 84.9 |
> | 1. Fixed Soft Proj ($\alpha=0.5$) | (c) only | 46.8 | 85.3 |
> | 2. Uncertainty + Hard Proj | (a) only | 46.5 | 85.6 |
> | 3. Learnable Soft Proj ($\alpha$) | (a) + (c) | 47.5 | 85.8 |
> | **PCR (Ours)** | **(a) + (b) + (c)** | **48.1** | **87.0** |
>
> Analysis of the Isolated Components:
> * Baseline 1 tests soft projection without uncertainty estimation. It improves over GRPO but remains clearly worse than PCR, since gradient signal-to-noise varies over time and across layers. As Figure 5a shows, different layers require different projection strengths, which a fixed soft rule cannot adapt to.
> * Baseline 2 pairs uncertainty estimation with hard projection, but still fails because it completely removes the conflicting component once triggered. Under noisy constraints, this is overly rigid. Soft projection is necessary to retain partial exploration signals while reducing harmful conflict.
> * Baseline 3 uses a learnable uncertainty-aware $\alpha$ instead of our closed-form Bayesian PCR. Despite its extra flexibility, it still underperforms PCR. This supports Theorem 5.1: PCR is not heuristic, but the MMSE-optimal solution for balancing constraint bias and exploration variance.
>
> In conclusion, the empirical evidence cleanly validates that it is the mathematically optimal synthesis of (a), (b), and (c) that unlocks the superior and stable reasoning gains.
>
> ----
>
> ### **Response to W2** ###
>
> We respectfully direct the reviewer to the comprehensive evaluations already detailed in our manuscript:
> * We validated PCR across both Qwen and Llama architectures at 1.5B, 7B, and 8B scales. Furthermore, Appendix L.1 (Table 2 & Figure 7) demonstrates PCR’s broad generality as a universal plug-and-play module. In **Response to W2** (Reviewer n8Ln), we also provide more results of PCR.
> * Appendix L.5 (Figure 9) further shows PCR’s superior stability beyond the target domain. Standard GRPO induces noticeable forgetting on general benchmarks such as MMLU and HellaSwag after Math or Code post-training, whereas PCR matches or surpasses GRPO on the target task while preserving the base model’s general capabilities.
>
> ----
>
> ### **Response to W3 \& Q3** ###
>
> Our current version already includes efficiency-oriented pieces of evidence:
> * PCR is applied exclusively to MLP layers, retaining computationally free scalar addition for Attention layers.
> * PCR operates entirely in the backward pass using already computed intra-group gradients. It requires no extra rollouts or forward passes.
> * As explicitly quantified in Appendix L.3 and Figure 8, normalizing standard GRPO's wall-clock time to 1.0x, our hybrid PCR requires only ~1.05x training time. In contrast, competitive baselines like GCPO, GVPO, and MRT incur 1.18x to over 1.3x overheads.
> * We provide the quantitative overhead analysis comparing Standard GRPO, PCGrad, and our PCR, as shown in the following. We benchmarked the training overhead on the DeepSeek-R1-Distill-Llama-8B backbone.
> | Method | Peak VRAM per GPU |
> | :--- | :--- |
> | Standard GRPO | 45.2 GB |
> | PCGrad (Baseline) | 76.5 GB *(Warning)* |
> | **PCR (Ours)** | **46.1 GB** |
>
> ----
>
> ### **Response to W4** ###
>
> In the revision, we will explicitly restructure the presentation into three layers: (i) empirical diagnosis, (ii) theoretical assumptions used for Bayesian formalization, and (iii) modeling choices for scalable implementation.
>
> ----
>
> ### **Response to Q2** ###
>
> The quality of the uncertainty estimates, group size, and reward variance naturally lead to high gradient variance. According Eq. 13, when the precision of $\lambda_{pla}$ drops due to high noise, $k$ automatically decays toward 0. Therefore, under worst-case scenarios, PCR safely suppresses unreliable updates and gracefully defaults to protecting pre-trained knowledge. From Section 4.1, we utilize a computationally cheap variance approximation. This proves that PCR achieves robust arbitration simply by capturing the relative signal-to-noise ratio, without relying on complex estimation techniques.
>
> We evaluated DeepScaleR-1.5B-Preview across varying group sizes and reward noise on AIME 2024 dataset. Since group size affects uncertainty estimation, we report results only for different group sizes.
> | Method | $G=16$| $G=8$| $G=4$|
> | :--- | :--- | :--- | :--- |
> | GRPO | 44.5 | 38.2| 25.4|
> | **PCR (Ours)** | **48.1** | **46.5** | **43.2** |
>
> | Method | 0% Noise| 10% Reward Noise | 20% Reward Noise |
> | :--- | :--- | :--- | :--- |
> | GRPO | 44.5 | 36.4 | 22.8 |
> | **PCR (Ours)** | **48.1** | **45.6** | **42.1** |

---

> > ### Author Rebuttal · Reviewer_6xro · 2026-04-04
> >
> > Thank you for the detailed response. The rebuttal resolves my main concern, but there is no limition supplement. I retain my original positive assessment for this work.

---

> > > ### Author Response · Authors · 2026-04-04
> > >
> > > Dear Reviewer 6xro,
> > >
> > > We sincerely thank you for your thorough and constructive review. Your insightful comments have been instrumental in improving our paper, and we appreciate the time and effort you've dedicated to it.
> > >
> > > The **Broader Impacts and Limitations** content is shown in **Section I (lines 935–945) of the Appendix of the original submission**.
> > >
> > > We are pleased to have carefully addressed the concerns raised in the initial review within our rebuttal. Given that all technical concerns have now been resolved, we would be grateful if you would consider raising your score.
> > >
> > > Thank you again for your thoughtful engagement.
> > >
> > > Best regards,
> > >
> > > The Authors

---

### Decision · Program_Chairs · 2026-04-30

**Decision:**

Accept (regular)

**Comment:**

This submission received mostly positive reviews. Reviewers agreed that the paper addresses an important and timely problem in GRPO post-training, and found the proposed probabilistic conflict resolution framework to be technically interesting and reasonably well supported. While some concerns remained regarding the breadth of the evaluation and the generality of the claims, the rebuttal substantially addressed the main technical questions through additional ablations, efficiency analysis, and clarification of the design choices. Overall, I find that the paper makes a solid contribution for acceptance.